

# Numerical Simulation of Present-day Kinematics at the Northeastern Margin of the Tibetan Plateau

Liming Li[1,2], Xianrui Li[3], Fanyan Yang[2], Lili Pan[4], Jingxiong Tian[2]

[1] School of Earth Science and Resources, China University of Geosciences (Beijing), Beijing 100083, China.

[2] Ningxia Institute of Geological Survey, Yinchuan, Ningxia 750021, China.

[3] Department of Earth and Space Sciences, Southern University of Science and Technology, Shenzhen 518055, China.

[4] School of Earth Sciences and Engineering, Sun Yat-Sen University, Zhuhai 519082, China.

*Corresponding to:* Liming Li (cugliming@cug.edu.cn)

**Abstract.** The slip rates of active faults at the northeastern margin of the Tibetan Plateau (NETP) must be clarified to understand the lateral expansion of the Tibetan Plateau and assess the seismic hazards in this region. To obtain the continuous slip rates of active faults at the NETP, we constructed a three-dimensional geomechanics-numerical model of the NETP. The model explains the fault systems, topographic undulations, and crustal stratigraphy of the study area. It also accounts for the physical rock properties, gravity fields, fault friction coefficients, initial crustal stresses, and
boundary conditions. The horizontal and vertical crustal velocities and slip rates of active faults in the study area were obtained from simulations using the aforementioned model. The results were then validated against independent geographic datasets. Based on the analysis of the fault kinematics in the study area, the Laohushan, middle–southern Liupanshan, and Guguan–Baoji faults, as well as the locked fault zone at the junction of the Maxianshan and Zhuanglanghe faults, represent potential hazard areas for strong earthquakes. However, as these faults are currently in
the stress accumulation stage, they are unlikely to cause a major earthquake in the short term. In contrast, it is likely that the Jinqiangshan–Maomaoshan fault will generate a ~$M_S7.0$ earthquake in the coming decades. Based on the analysis of several profiles across the NETP, the deformations at the NETP are continuous in the Bayan Har and Qaidam blocks, as well as in the block-like in Qilian Block, particularly around the Haiyuan Fault.

## 1 Introduction

The northeastern margin of the Tibetan Plateau (NETP) is the growth front of the Tibetan Plateau (TP) system and a modern topographic and tectonic framework that formed due to the expansion of the TP toward its periphery, which has been ongoing since the Indian and Eurasian plates collided (Zhang et al., 2013; P. Zhang et al., 2014). This region contains some of the most easily deformed and flowing crustal matter worldwide, with extremely intense tectonic movements and seismic activities, and a complex and highly varied tectonic system characterized by strong Cenozoic
deformation (Zhang, 1999). The NETP is intersected by many deep and large faults, such as the Haiyuan, West Qinling, Huanghe, Luoshan, Yunwushan, and Xiaoguanshan faults, which divide the NETP into several active tectonic blocks. These blocks include the Alxa, Ordos, Qilian, Qaidam, and Bayan Har blocks (Zhang et al., 2003; see Figure 1). The faults have been highly active since the Holocene and have caused many large earthquakes, including 17 earthquakes



with a magnitude of ≥ 7 and several M ≥ 8.0 earthquakes, such as the 1654 M8.0 Tianshui, 1739 M8.0 Pingluo, 1879

M8.0 Wudu, 1920 M8.0 Haiyuan, and 1927 M8.0 Gulang earthquakes (Figure 1). The generation, occurrence, and strength of an earthquake are closely related to fault activity. The segmentation of active faults and their long-term slip rates are imperative in seismological research and important predictors of medium- and long-term seismic hazards (Ding et al., 1993; Xu et al., 2018). Accurate fault slip rates can calculate seismic cycles (Shen et al., 2009) and assess the seismogenic potential (Bai et al., 2018; Hergert et al., 2010). Continuous fault slip rates can also be used to

reconstruct the tectonic evolution of an area, which may provide important insights into the lateral expansion and mechanical deformation mechanisms of the TP (Royden et al., 1997; Tapponnier et al., 1982; Zhang et al., 2004). Therefore, detailed studies must be performed on the slip rates of active faults in the NETP.

The slip rates of active faults in the NETP have been extensively studied using geologic (Chen et al., 2019; Li et al., 2009; Li et al., 2018; Matrau et al., 2019; Wang et al., 2021) and geodetic (Hao et al., 2021; Li et al., 2019; Li, Pierce,

et al., 2021; X. Li et al., 2017) approaches. However, both approaches have limitations. The fault slip rates obtained from geologic approaches only represent the rate at each point of measurement, which does not necessarily represent the fault as a whole. Furthermore, geologic fault slip rates are generally averaged over long periods of time and therefore provide limited information regarding present-day fault movements. Conversely, the geodetic approach assumes that each block is rigid with negligible internal deformation. The results of several studies demonstrated that

the internal deformations of the TP are "continuous" (Royden et al., 1997; Zhang et al., 2004). Therefore, internal block deformation in the NETP cannot be ignored. Numerical simulations are a powerful tool for systematic fault kinematics studies (Hergert et al., 2011; Li, Hergert, et al., 2021) as they provide a comprehensive view of current fault activities. However, the focus of previous numerical studies on the NETP has been placed primarily on the crustal stress environment and seismic activity of the NETP (Pang et al., 2019b; Sun et al., 2018, 2019) or on the analysis of

factors affecting crustal movements (Pang et al., 2019a; Zhu et al., 2018). In contrast, detailed studies of the three-dimensional (3D) kinematics of the crust and active faults in the NETP are scarce.

In this study, we constructed a 3D geomechanics model of the NETP fault system to elucidate the slip rates of the major active faults, crustal deformation characteristics, and partitioning of deformation modes (between block-like and continuous deformation). The results were then used to analyze the seismic hazards of major faults at the NETP.

Based on this numerical study, important data were obtained that provide insights into the motions and transformations of active tectonic structures at the NETP.



**Figure 1.** Map of active faults and earthquakes of the NETP. Black lines represent the active faults. faults discussed in the text are labeled as followed: F1-1 = LLLF = Lenglongling fault; F1-2 = JQHF = Jinqinaghe fault; F1-3 = MMSF = Maomaoshan fault; F1-4 = LHSF = Laohushan Fault; F1-5 = HYF = Haiyuan Fault. F1-6 = LPSF = Liupanshan Fault; F2-1 = ZZSF = Zhuozishan Fault; F2-2 = HHF = Huanghe Fault; F2-3 = LSF = Luoshan fault; F2-4 = YWSF = Yunwushan fault; F2-5 = XGSF = Xiaoguanshan Fault; F3-1 = DTH-LXF = Daotanghe–Linxia fault; F3-2 = WQLF = West Qinling Fault. F4-1 = EKLF = Eastern Kunlun fault; F4-2 = TZF = Tazang Fault; F5 = TJSF = Tianjingshan fault; F6 = WHLSF = West Helanshan fault; F7 = NSSF = Niushoushan fault; F8 = EHLSF = East Helanshan fault; F9 = ZYGF = Zhengyiguan fault; F10 = LHTF = Lvhuatai fault; F11 = YCF = Yinchuan fault; F12 = YTSF = Yantongshan fault; F13 = QSHF = Qingshuihe fault; F14-1 = ZLSF = Zhuanglanghe fault; F14-2 = MXSF=Maxianshan Fault. F15-1 = WLJSF = West section of Lajishan fault; F15-2 = ELJSF = East section of Lajishan fault; F16-1 = DB-BLJF = Diebu–Bailongjiang fault; F16-2 = WD-KXF = Wudu–Kangxian fault.





## 2 Model concept and input

### 2.1 Model geometry

The 3D geomechanics model of this work is a rectangular cuboid with an E–W length of 654 km (101°E–108°E), N–S length of 777 km (33°N–40°N), and thickness of 80 km. The topographic undulations of the model's surface were characterized using GTOPO30 elevation data, which has a resolution of 30 arcseconds. The model comprises four layers: the upper crust, middle crust, lower crust, and upper mantle, from top to bottom. The geometric data of the layer interfaces were derived from CRUST1.0 (Laske et al., 2013).

Based on the cutting depth, the faults of the model can be categorized into lithospheric and intracrustal faults. Lithospheric faults (e.g., F1, F2, F3, and F4) are plate boundaries because they cut through the Moho and reach the bottom of the model (Figure 2a and Figure 2b; Table 1). All other faults are intracrustal faults because the depth data indicate they terminate in the upper, middle, or lower crust. The fault traces were obtained from Xu et al. (2016) and the attitudes were derived from previously published data including surface-based fault surveys and deep seismic sounding profiles (Table 1). The results of many studies indicated a low-velocity body in the upper crust and that the deformations in the upper crust are not coupled to those in the underlying crust (Bao et al., 2013; Wang et al., 2018; Ye et al., 2016). Therefore, the upper and middle crusts were decoupled in the model to allow the upper crust to slide freely along its bottom interface.

The model was meshed using tetrahedron elements. The elements were 1–2 km wide at the faults. The largest elements outside the faults were ~10 km wide. The model contained 8,463,583 elements (Figure 2a).

### 2.2 Rock properties

The stress–strain relationships of the rocks were modeled using their elastic parameters (Hergert et al., 2010, 2011; Li, Hergert, et al., 2021). The model was divided into three major tectonic elements—the Alxa, Ordos, and TP blocks. The TP block comprises the Qilian, Qaidam, and Bayan Har blocks. Each tectonic element contained the elastic parameters of the upper, middle, and lower crust as well as those of the upper mantle, with each set of elastic parameters including the Young's modulus, density, and Poisson's ratio (Figure 2a). The density and Poisson's ratio were derived from CRUST1.0 (Laske et al., 2013). The Young's modulus was calculated based on the P-wave velocities, S-wave velocities, and densities derived from CRUST1.0 using the empirical equation of Brocher (2005). The Young's modulus, computed from seismic wave velocities, corresponded to the dynamic elastic modulus, which is generally greater than a rock's static Young's modulus. Therefore, the Young's modulus values were converted into static Young's moduli using the empirical equation of Brotons et al. (2016). The elastic parameters used in the model are listed in Table 2.

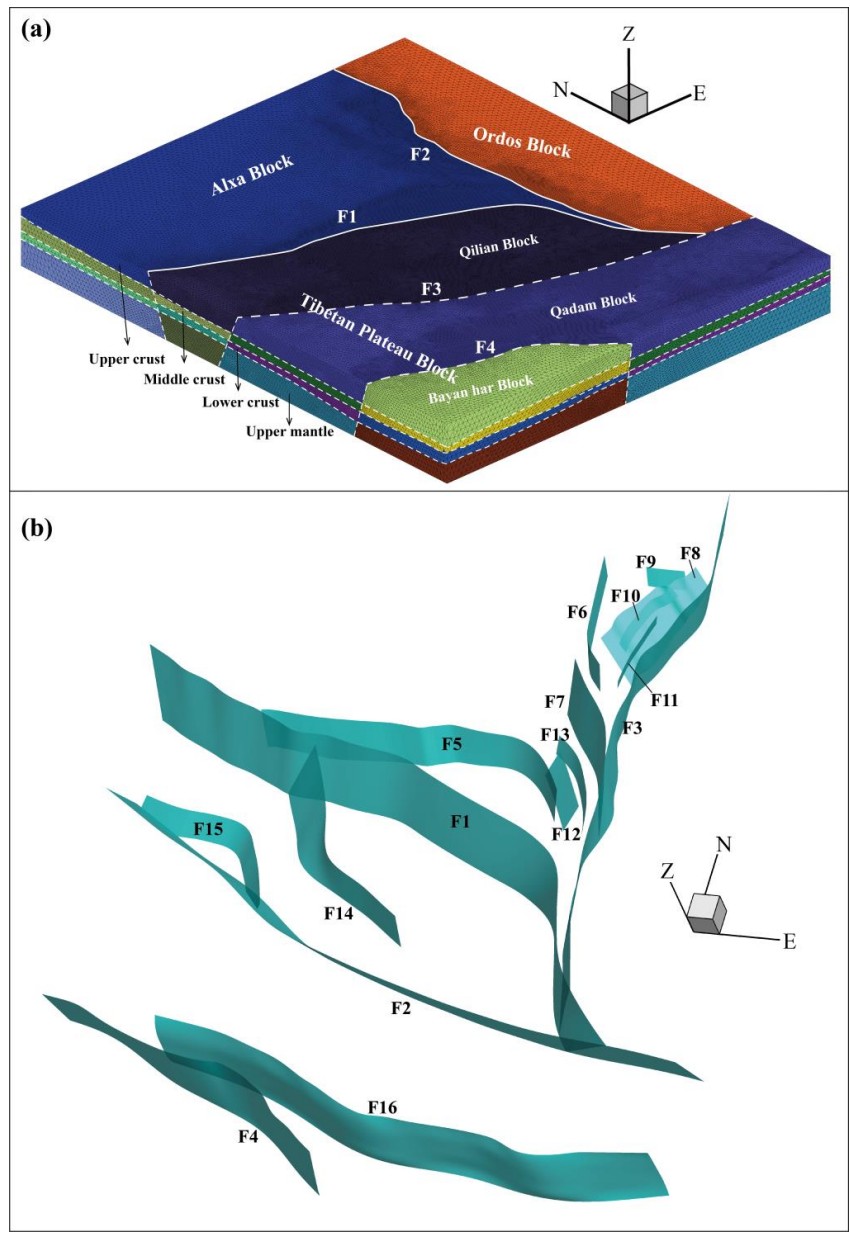

**Figure 2. Model geometry and implemented fault system. (a) Distribution of the rock properties employed for the model. Different colors represent different rock properties. (b) Cyan surfaces indicate the faults implemented in the model as frictional contact surfaces. Fault names can be found in Figure 1.**

105



110

**Table 1.** Geometric parameters of faults in the model

|  | Fault name | Strike | Dip direction (°) | Dip (°) |
|---|---|---|---|---|
| F1 | / | NWW-SSE | SW | 70 |
| F2-1 | ZZSF | N-S | W | 70 |
| F2-2 | HHF | N-S | W | 70 |
| F2-3 | LSF | N-S | W | 80 |
| F2-4 | YWSF | N-S | W | 70 |
| F2-5 | XGSF | N-S | W | 70 |
| F3-1 | DTH-LXF | NWW-SSE | NE | 70 |
| F3-2 | WQLF | | | |
| F4-1 | EKLF | NW-SE | NE | 75 |
| F4-2 | TZF | | | |
| F5 | East section of TJSF | NW-SE | SW | 70 |
| | West section of TJSF | E-W | S | 70 |
| F6 | WHLSF | N-S | W | 80 |
| F7 | NSSF | NW-SE | SW | 70 |
| F8 | EHLSF | NE-SW | SE | 60 |
| F9 | ZYGF | E-W | S | 60 |
| F10 | LHTF | NNE-SSW | SE | 70 |
| F11 | YCF | NNE-SSW | NW | 70 |
| F12 | YTSF | NW-SE | SW | 65 |
| F13 | QSHF | NW-SE | SW | 45 |
| F14-1 | ZLHF | NNW-SSE | SW | 45 |
| F14-2 | MXSF | NW-SE | SW | 80 |
| F15 | LJSF | NWW-SSE | SW | 50 |
| F16-1 | DB-BLJF | NW-SE | SW | 70 |
| F16-2 | WD-KXF | E-W | SW | 70 |

The detailed fault names are defined in Figure 1.

**Table 2.** Material parameters of the finite element model

| | Alxa Block | | | Ordos Block | | | Tibetan Plateau Block | | |
|---|---|---|---|---|---|---|---|---|---|
| | E(Gpa) | $\rho$ (g/cm$^3$) | $\nu$ | E(Gpa) | $\rho$ (g/cm$^3$) | $\nu$ | E(Gpa) | $\rho$ (g/cm$^3$) | $\nu$ |
| Upper crust | 77.4 | 0.244 | 2.74 | 77.4 | 0.244 | 2.74 | 76.8 | 0.243 | 2.74 |
| Middle crust | 84.2 | 0.247 | 2.78 | 84.2 | 0.247 | 2.78 | 84.5 | 0.246 | 2.78 |
| Lower crust | 113.0 | 0.259 | 2.95 | 113.0 | 0.259 | 2.95 | 110.0 | 0.257 | 2.93 |
| Upper mantle | 194.0 | 0.278 | 3.39 | 191.0 | 0.278 | 3.37 | 187.0 | 0.278 | 3.36 |

E, $\rho$, and $\nu$ are Young's modulus, density, and Poison's ratio, respectively.


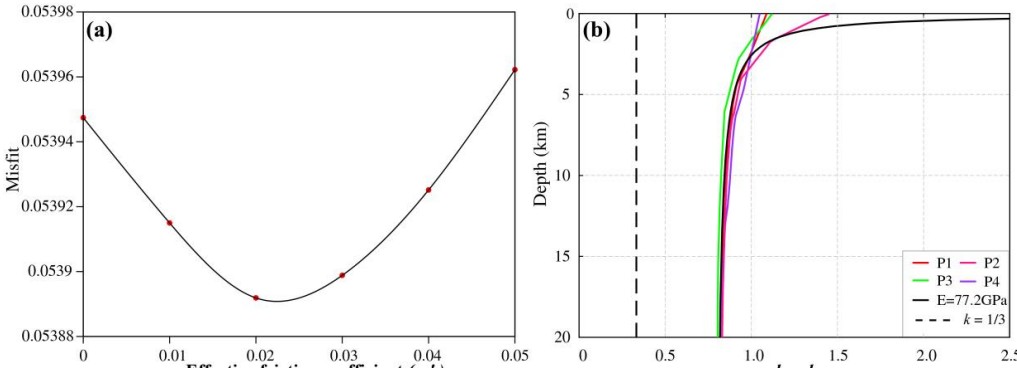

**Figure 3. (a) Velocity misfit of the crustal velocity between the modeled results and GPS measurements as a function of the effective friction coefficient. The optimal friction coefficient was determined to be 0.02. (b) Depth profiles of the initial k-values at four test sites indicated in Figure 1. The solid black line shows the stress state based on Equation (3), which was used as a reference stress state. For comparison, the low k-value based on the uniaxial strain state is also shown (dashed black line).**

**2.3 Friction coefficient**

In our model, frictional contacts between fault surfaces were considered obeying the Mohr–Coulomb friction law:

$$\sigma_s = C_0 + \mu \cdot (\sigma_n - P_f) = C_0 + \mu' \cdot \sigma_n, \tag{1}$$

where $\sigma_s$ is the shear force on the fault surface, $C_0$ is the cohesion, $\mu$ is the coefficient of friction, $\sigma_n$ is the normal stress on the fault surface, $P_f$ is the pore pressure, and $\mu'$ is the effective coefficient of friction of the fault surface when accounting for the pore pressure. In our model, the cohesion $C_0$ of the rocks was assumed to be negligible (Jamison et al., 1980).

The frictional relations of the fault surface are critically important for the kinematics of a fault. However, these relations can be complex in a fault and also vary in time and space. The exact magnitude of the friction coefficient is affected by various external factors including fluid seepage, temperature variations, stress states, and movement. Therefore, it is challenging to obtain the precise friction coefficient of a fault. However, the results of many studies showed that large strike-slip faults generally have low effective friction coefficients. For instance, the Haiyuan Fault has friction coefficients as low as 0.05 (He et al., 2013) and faults on the eastern margin of the TP have friction coefficients as low as 0.02 (Li et al., 2015; Li, Hergert, et al., 2021). Because of this information, simulations were performed with a series of friction coefficients (0–0.05) and the results were compared with GPS observations (Cianetti et al., 2001). The results showed that setting a friction coefficient of 0.02 for all faults yields the smallest fitting error (0.05389; Figure 3a). To minimize the fitting error, localized adjustments were made for the friction coefficients of the F1 and F3 faults, which are large strike-slip faults. The final friction coefficients of F1, F3, and all other faults were 0.01, 0.1, and 0.02, respectively, with a fitting error of 0.0446.

**2.4 Initial stress state**

The initial stress affects the stress state of a fault, which controls the kinematic state of the fault via the friction coefficient. Therefore, the selection of appropriate initial stresses is important when performing numerical simulations





based on geomechanics models. The initial stress model that is most commonly employed in numerical studies of the TP is the uniaxial strain reference state (Sun et al., 2019; Zhu et al., 2016), which predicts that all deformations due to gravitational loading occur in the vertical direction and that no expansion or contraction occurs in the lateral direction. In this stress state, the ratio ($k$) of mean horizontal stress to vertical stress is only dependent on the Poisson's ratio:

$$k = \frac{(S_H + S_h)}{2S_V} = \frac{v}{1-v}, \quad (2)$$

where $S_H$, $S_h$, and $S_V$ are the maximum horizontal, minimum horizontal, and vertical stress, respectively, and $v$ is Poisson's ratio. For the typical $v$-value of 0.25, Equation (2) gives $k = 1/3$, which implies that the vertical stresses acting on the rock mass far exceed the horizontal stresses and that the crust is always in a normal faulting or extensional stress regime. However, this assumption contradicts the thrust and strike-slip stress regimes common in the crust. Furthermore, $k$-values obtained globally from *in situ* measurements always greatly exceed 1/3 (Hergert and Heidbach,

150 2011).

Based on a spherical shell model, Sheorey (1994) proposed a method for the estimation of the initial crustal stress, which accounts for the curvature of the Earth, the properties of crustal and mantle materials, temperature fields, and other thermally dependent properties, as shown in Equation (3):

$$k = 0.25 + 7E \cdot \left(0.001 + \frac{1}{z}\right), \quad (3)$$

Where $E$ is the Young's modulus of the rock (Gpa); and $z$ is the depth from the surface (m). Because the $k$-values obtained by this method are generally consistent with those obtained from ultra-deep boreholes (Hergert and Heidbach, 2011), they are often used in numerical studies performed by researchers outside China (Buchmann et al., 2007; Rajabi et al., 2017; Reiter et al., 2014). However, this approach is rarely used in China. Equation (3) is used in our model to calculate the initial stresses. The exact methods for obtaining the initial stresses proposed by Sheorey (1994) have

been described previously by Hergert (2009). Figure 3b shows that the initial stresses in our model agree with the theoretical results given by Sheorey (1994).

## 2.5 Kinematic boundary conditions

The GPS velocity field data of Wang et al. (2020) were used as lateral boundary conditions for our model. Because of the scarcity of local vertical deformation data, the model boundaries were only constrained in the horizontal

direction and it was assumed that the lateral velocities of the 3D model do not vary with depth. The vertical displacements of the model were unconstrained. The top surface of the model was configured as a free boundary, whereas the bottom surface slid freely in the horizontal direction, with a vertical velocity of 0. The detailed boundary conditions are shown in Figure 4.

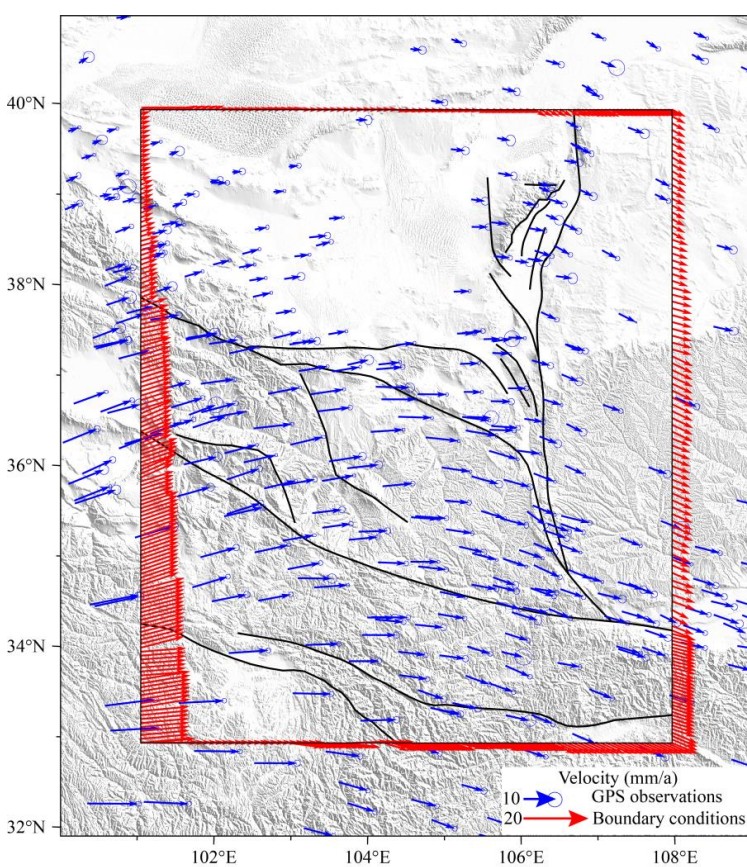

**Figure 4. Boundary conditions of the model. The blue arrows represent the GPS observation velocities according to Wang et al. (2020). The red arrows at the boundary represent the boundary velocities calculated by the interpolation of the GPS observations.**

## 3 Results

### 3.1 Horizontal crustal velocities

The distribution of horizontal crustal velocities in the study area is shown in Figure 5. In terms of the direction, the western and central parts of the study area are moving in the NEE and near-EW directions and the eastern part gradually changes in motion toward the SW or SEE directions. Therefore, the study area is characterized by clockwise crustal motions. In terms of crustal speed, the model shows that the crustal speeds in the southwestern part of the study area are high, whereas they are low in the northeastern part. The Alxa Block in the north and Ordos Block in the east

have crustal speeds of ~4–6 mm/a, indicating that their internal deformation is low. Therefore, these blocks are relatively stable. The Bayan Har Block in the south has the highest crustal speed (up to 13–14 mm/a). The Qaidam Block exhibits crustal speeds of 11–12 mm/a on its western side, decreasing to 9 mm/a on the eastern side. The western part of the Qilian Block has a crustal speed of 9–10 mm/a, decreasing to 8 mm/a on the southeastern side.




Figure 5 shows that large strike-slip faults control the distribution of crustal velocities. For example, the F1, F3, and
F4 faults vertically cut through the model and act as "separators" regarding the crustal velocity distribution. This
phenomenon is most pronounced at the F1 fault. The crustal speed on the southern side of the Qilian Block is 9–11
mm/a, whereas that on the northern side of the Alxa Block is 4–6 mm/a. Therefore, the crustal speeds on opposing
sides of the F1 fault can differ by ≥3 mm/a.

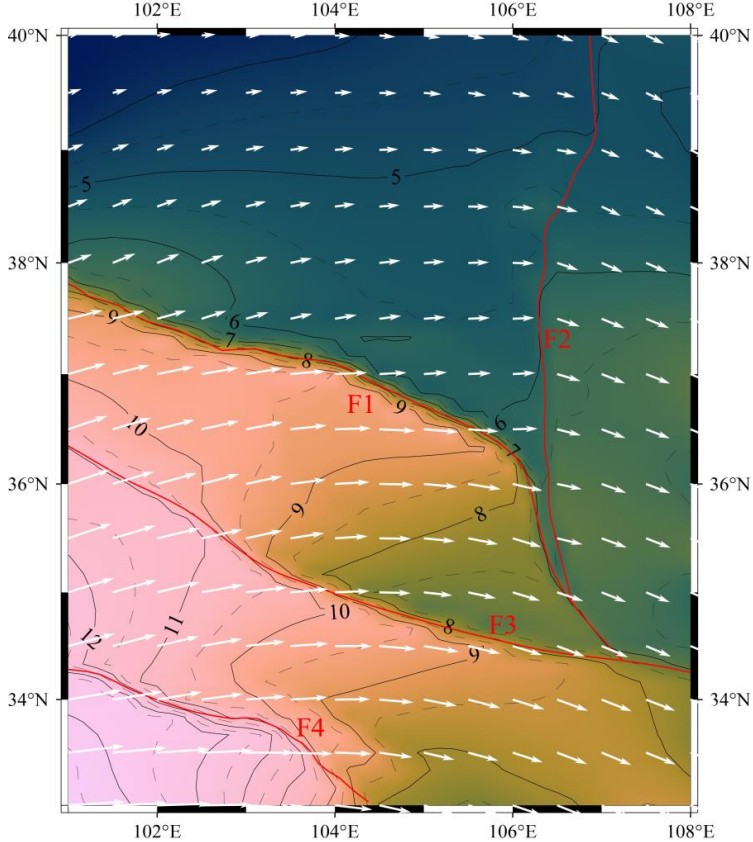

**Figure 5. Distribution of modeled crustal surface velocities of the NETP with a grid interval of 0.5° both in longitude and
latitude. White arrows represent the crust movement direction. The background color contours represent the magnitude
of the velocity in mm/a. The red lines represent the faults implemented in the model. The names are defined in Figure 1.**

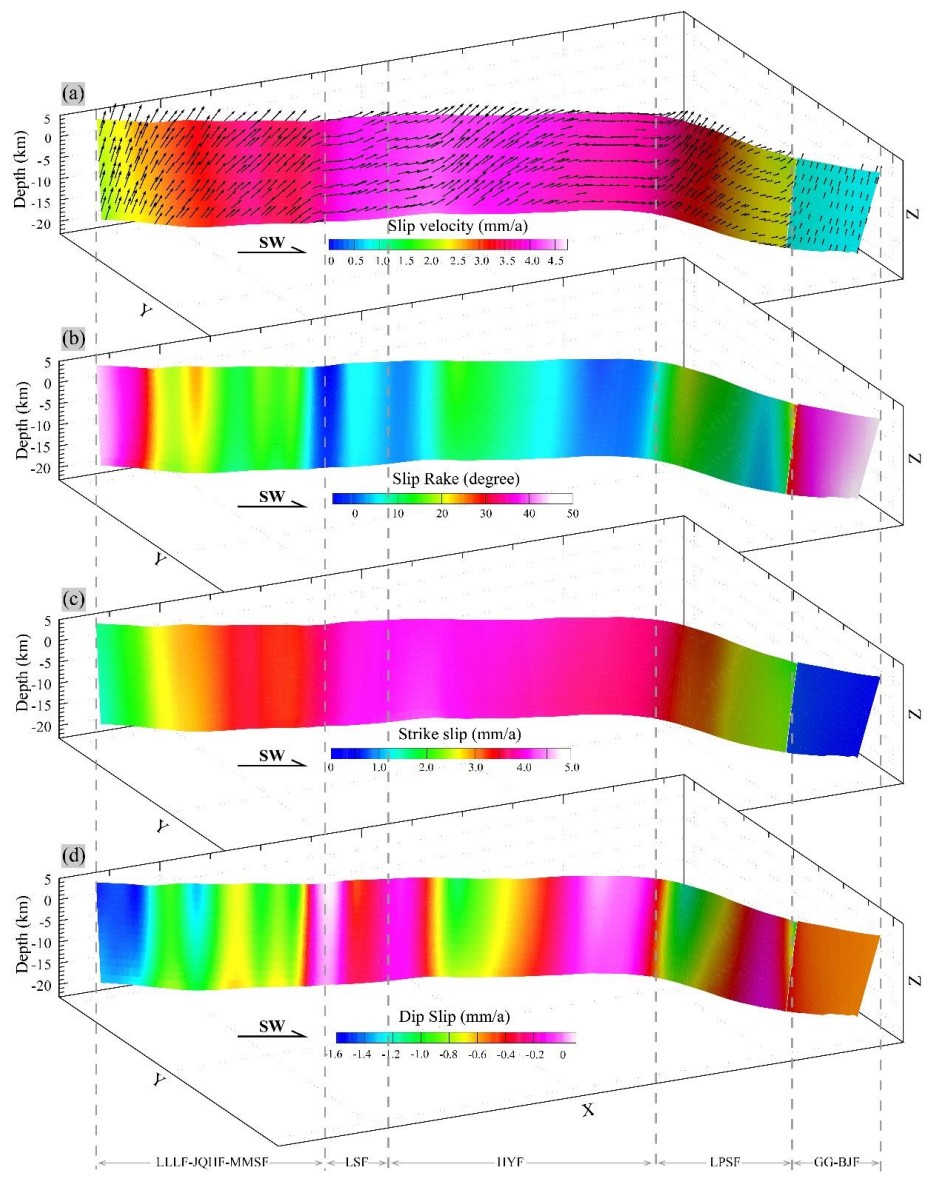

**Figure 6. Distribution of the modeled fault slip rates on the F1 fault. View from NE to SW. The ratio of the vertical to horizontal scale is 4:1. (a) Total slip rates with slip directions (black arrows). Parts of the fault has an oblique-slip component. (b) Slip rake. Positive values indicate that the fault's movement has a thrust component. (c) Slip rates along the fault's strike. Positive values represent sinistral strike-slip. (d) Slip rates along the fault's dip direction. Negative values represent thrust faulting.**






### 3.2 Fault slip rates of the main faults

The dominant depth range of seismic activities in the study area is 5–15 km, with a few events at depths up to 20 km (Li et al., 2020). Therefore, we extracted the kinematic characteristics up to a depth of 20 km for the main faults of the study area using our 3D geomechanics model. The extracted kinematic characteristics are described below.

#### 3.2.1 Slip rate of the F1 fault

From west to east, the F1 fault comprises the Lenglongling, Jinqianghe, Maomaoshan, Laohushan, Haiyuan, Liupanshan, and Guguan–Baoji faults (Figure 1). Two earthquakes with a magnitude M ≥ 8.0 have occurred along this fault—the 1920 M8.0 Haiyuan Earthquake and 1927 M8.0 Gulang Earthquake, which formed 220 and 120 km long surface rupture zones, respectively (Guo et al., 2020; Liu-Zeng et al., 2015; Zhang et al., 1987). The 3D kinematic state of the F1 fault was extracted from the 3D geomechanics model, as shown in Figure 6. On the western part of the F1 fault (Lenglongling, Jinqianghe, and Maomaoshan faults), the slip rates increase from 2.0 to 4.0 mm/a from west to east and continue to increase further east (Laohushan Fault and western part of the Haiyuan Fault) up to a maximum of 4.5 mm/a. The slip rates of the F1 fault decrease in the middle of the Haiyuan Fault. The eastern part of the Haiyuan Fault has a slip rate of ~3.5 mm/a. The slip rate of the Liupanshan Fault decreases in the SE direction and bottoms out at ~2.0 mm/a; at the Guguan–Baoji Fault, the slip rate is lower than 1 mm/a (Figure 6a). The slip rates obtained by our 3D geomechanics model for the Haiyuan Fault are much lower than older estimates (8.0–12 mm/a; Burchfiel et al., 1991; Lasserre et al., 1999; Zhang et al., 1988) but similar to more recent estimates (3.2–4.5 mm/a; Li et al., 2009; Matrau et al., 2019; Y. Li et al., 2017).

Although the F1 fault is predominantly a left-lateral strike-slip fault, it also has a thrust component (see Figure 6b). The Lenglongling, Jinqianghe, Maomaoshan, middle Haiyuan, and Liupanshan faults have the rake ranging from 10° to 20°, whereas the Guguan–Baoji Fault has the rake varying from 40° to 50°, indicating that they are oblique thrust faults. The Laohushan and western and eastern Haiyuan faults have rake below 10°, showing that left-lateral strike-slip faulting is dominant at these faults. Figures 6c and 6d show the continuous slip rates of the F1 fault along its strike and dip, respectively. The slip rates along the strike are like the total slip rates. The Laohushan and Haiyuan faults have the highest slip rates on the F1 fault. Conversely, the Lenglongling, Jinqianghe, Maomaoshan, and Liupanshan faults have high dip-slip rates.

#### 3.2.2 Slip rate of the F2 fault

The F2 fault comprises the Zhuozishan, Huanghe, Luoshan, and Yunwushan–Xiaoguanshan faults (Figure 1). Figure 7 shows that right-lateral strike-slip faulting is prevalent across the entire F2 fault. However, the magnitude of the dip-slip component varies from one location to another. The Zhuozishan Fault is an oblique-slip reverse fault with slip rates ranging from 0.8 to 1.6 mm/a. The Huanghe Fault has a slip rate of 1.6–2.6 mm/a. Its northern segment is an oblique-slip normal fault, whereas its southern segment has no dip-slip component. The Luoshan and Yunwushan faults have slip rates ranging from 2.6 to 3.0 mm/a and are dominated by right-lateral strike-slip faulting. The Xiaoguanshan Fault is an oblique-slip reverse fault with slip rates ranging from 3.0 mm/a (northern end) to 1.4 mm/a (southern end).


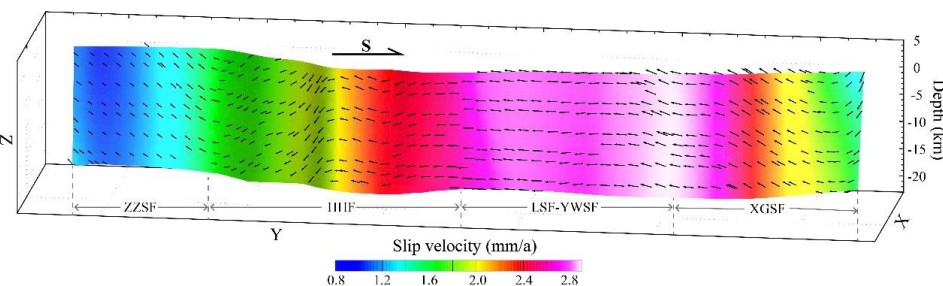

Slip velocity (mm/a)

0.8  1.2  1.6  2.0  2.4  2.8


**Figure 7. Distribution of the modeled fault slip rates on the F2 fault. View from north to south. The black arrows represent the slip directions. The ratio of the vertical to horizontal scale is 4:1.**

### 3.2.3 Slip rate of the F3 fault

The F3 fault includes the Daotanghe–Linxia and West Qinling faults (Figure 1) and is a uniform left-lateral strike-slip

fault, as shown in Figure 8. The western Daotanghe–Linxia Fault is an oblique-slip reverse fault, which has slip rates

ranging from 0.8 to 1.8 mm/a. The West Qinling Fault has slip rates varying from 1.8 to 2.8 mm/a, but each part of

this fault has slightly different kinematics. The West Qinling Fault is an oblique-slip reverse fault west of Zhangxian,

but is dominated by left-lateral strike-slip faulting with a small thrust component in the Zhangxian–Tianshui–Baoji

region. The West Qinling Fault is an oblique-slip normal fault only in the vicinity of Baoji.

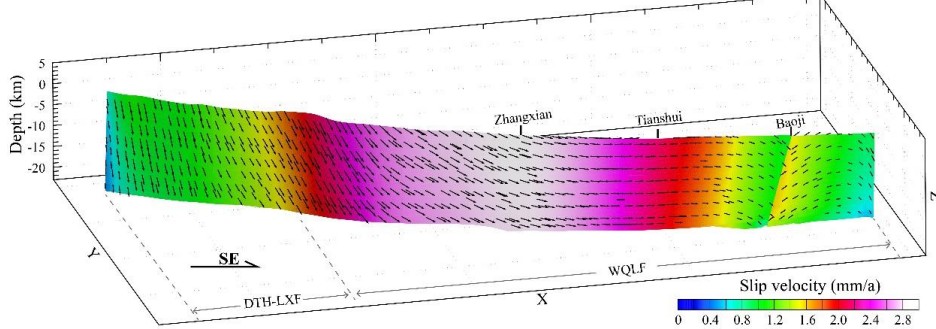

Slip velocity (mm/a)

0   0.4  0.8  1.2  1.6  2.0  2.4  2.8


**Figure 8. Distribution of the modeled fault slip rates on the F3 fault. View from NW to SE. The black arrows represent the slip directions. The ratio of the vertical to horizontal scale is 4:1.**

### 3.2.4 Slip rate of the F4 fault

The F4 fault in our model only includes a part of the Maqu segment of the East Kunlun Fault and a part of the Tazang

Fault. Several researchers reported slip rates ranging from 7.68 to 11.5 mm/a for the Maqu segment of the East Kunlun

and Tazang faults (J. Zhang et al., 2014; Van der Woerd et al., 1999). In contrast, others reported slip rates that steadily

decrease in the eastward direction—slip rates ranging from 3 to 5 mm/a in the Maqu segment (J. Li et al., 2016; Li,

2009) and from 1.4 to 3.2 mm/a for the Tazang Fault (Ren et al., 2013). Some researchers even reported slip rates

below 1 mm/a (Kirby et al., 2007). The 3D kinematics of the F4 fault indicate that the slip rates around Maqu only





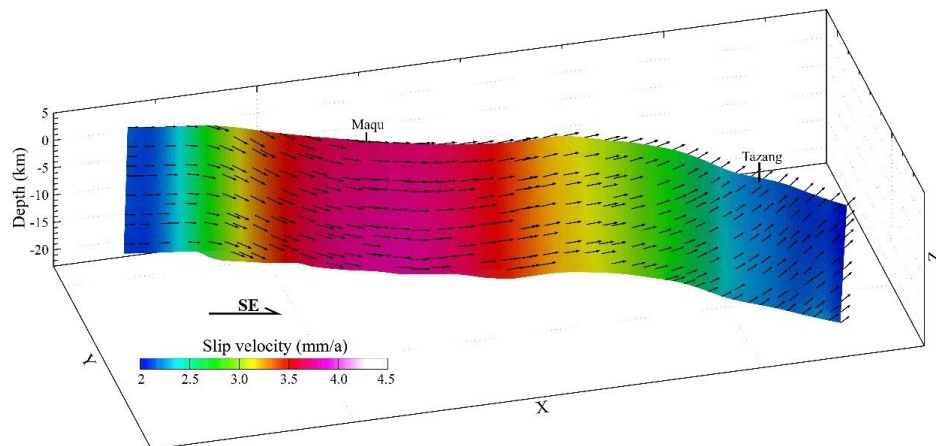


**Figure 9. Distribution of the modeled fault slip rates on the F4 fault. View from NW to SE. The black arrows represent the slip directions. The ratio of the vertical to horizontal scale is 4:1.**

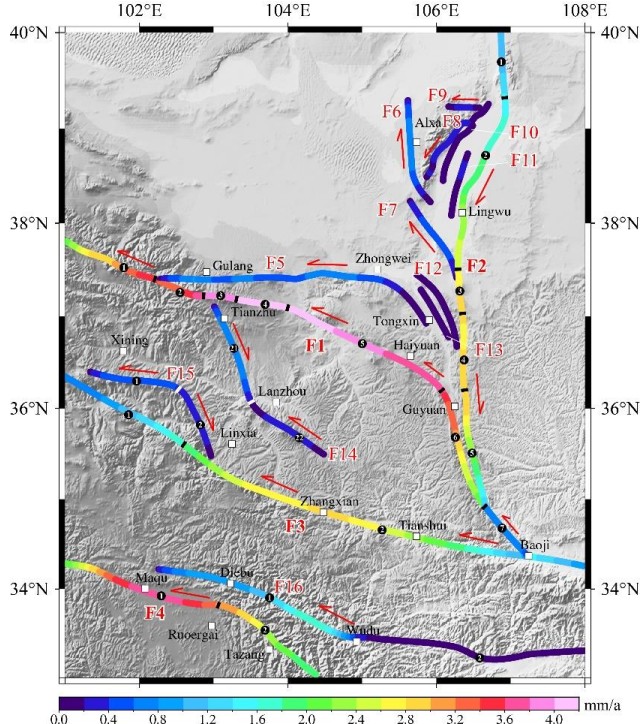

**Figure 10. Modeled horizontal fault slip rates and slip senses. The fault names are defined in Figure 1.**

range from 3.0 to 4.0 mm/a. Furthermore, the slip rates around eastern Tazang are as low as 2 mm/a (Figure 9). Hence, the eastern East Kunlun Fault does not retain the high slip rate of the western East Kunlun Fault. Figure 9 also shows that the F4 fault, which is a left-lateral strike-slip fault, exhibits thrust near the southeastern segment of the Tazang Fault.





The 3D kinematics of other faults in the study area are not described in this study. Their horizontal velocities are
listed in Figure 10.

### 3.3 Vertical velocities

Figure 11a shows the distribution of vertical velocities in the study area. In the study area, surface subsidence can be
observed only in the Yinchuan Basin west of Huanghe Fault, in the Ordos basin in the east, and in the Sikouzi Basin
in the southern Ningxia arc tectonic belt. Most of the subsidence rates range between 0 and 0.2 mm/a, except in the
center of the Yinchuan Basin, which exhibits subsidence rates varying from 0.2 to 0.4 mm/a. Based on paleomagnetic
studies, the Sikouzi Basin had a subsidence rate of 0.22 mm/a during the Pliocene (Wang et al., 2011), whereas the
subsidence rate of the Yinchuan Basin has been 0.32 mm/a since the Middle Pleistocene (Ma et al., 2021). Both values
are consistent with those derived from our simulation.

Most other parts of the study area exhibit uplifting, albeit at low rates (generally less than 0.2 mm/a). Most areas with
high uplift rates (0.8–1.0 mm/a) are in the Qilian Block such as the Lenglinglong Fault and Jinqianghe Fault, the
southern side of the middle Haiyuan Fault, the western side of the Liupanshan segment, and the southern side of the
Lajishan Fault (Figure 11a). Areas with high vertical fault velocities are also on the F1 and F3 thrust faults that form
the NS boundary of the Qilian Block (Figure 11b). The high vertical velocities of the Qilian Block are caused by the
tectonic setting. Because of the NE expansion of the TP and the Alxa Block under thrusting the tectonic transition
zone of the NETP, the Qilian Block is compressed from two directions (Ye et al., 2015). Therefore, outward thrust
stacking occurs on the southern and northern boundaries (F1 and F3 faults) of the Qilian Block.

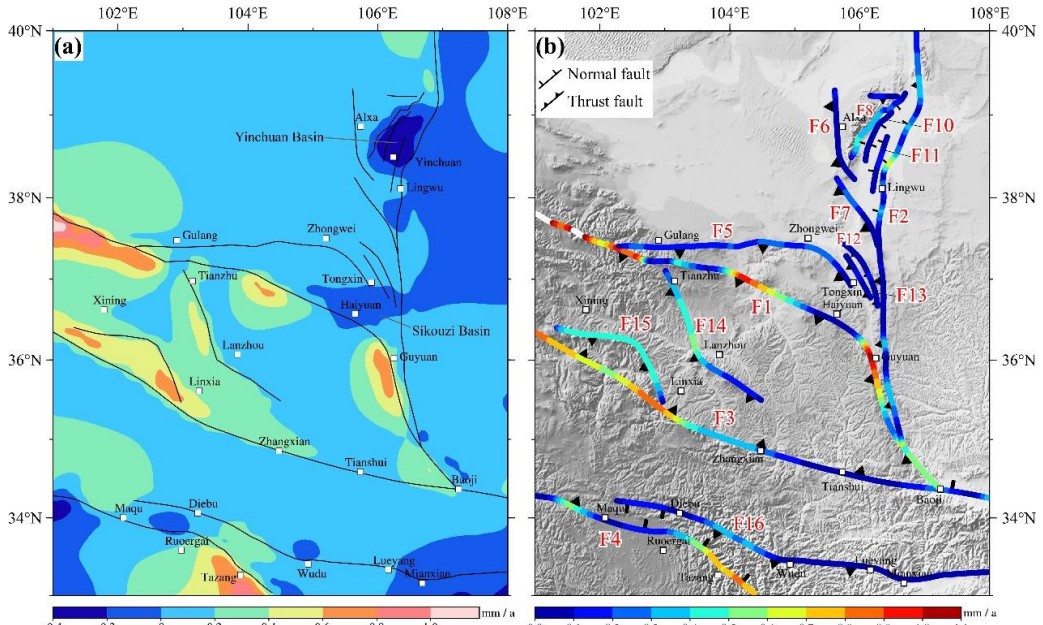

**Figure 11. (a) Modeled vertical velocity at the surface. Negative values indicate subsidence, whereas positive values represent uplift. (b) Modeled vertical slip rates on faults at the surface. The fault names are defined in Figure 1.**


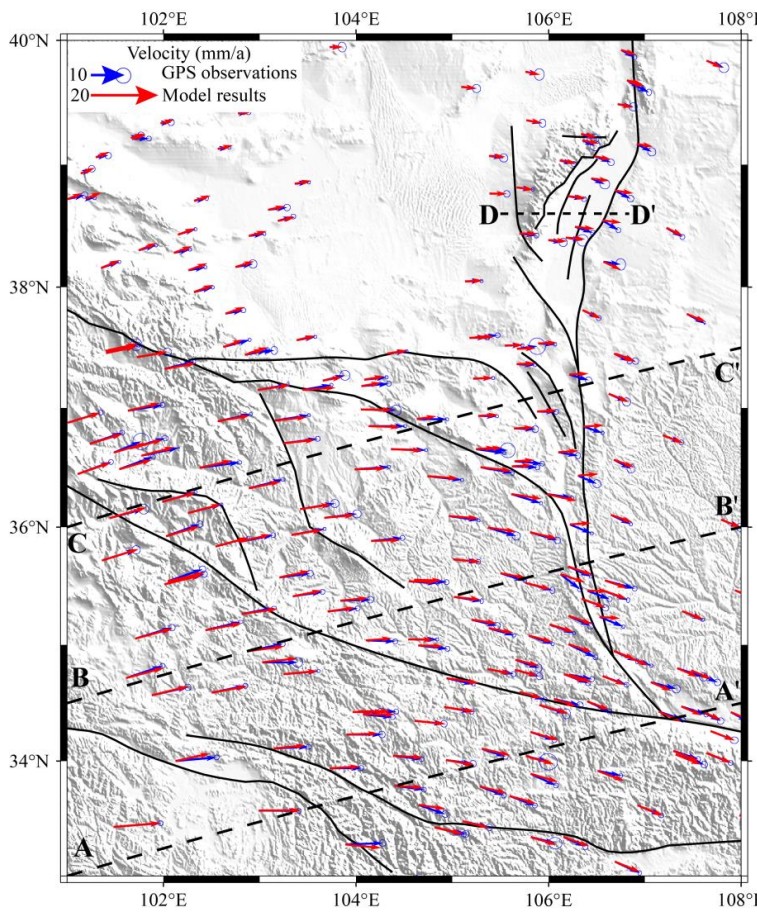

**Figure 12. Comparison of the modeled horizontal velocities and GPS velocities. The red arrows represent the modeled results and the blue arrows are the GPS measurements (Wang et al., 2020). The dashed line is the location of the profile in Figure 13 and Figure 15.**

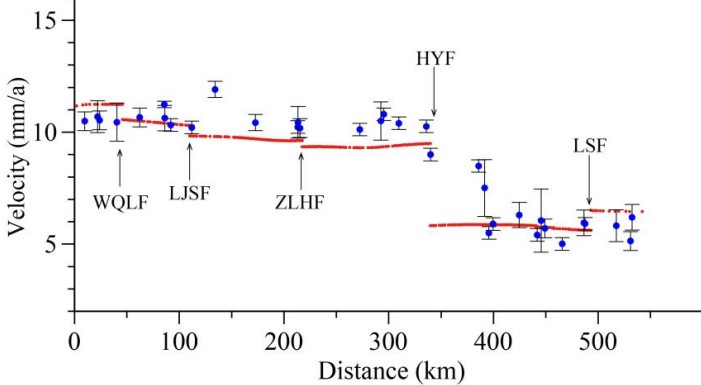


**Figure 13. Comparison of the modeled horizontal velocities and GPS velocities along the C-C' profile in Figure 12. The red points are model data and blue circles indicate GPS measurements. The fault names are defined in Figure 1.**



**Table 3. Comparison of model slip rates with geological slip rates**

| Fault name | | Modeled rate (mm/a) | | Geological rate (mm/a) | | References |
|---|---|---|---|---|---|---|
| | | Lateral [a] | Vertical [b] | Lateral [a] | Vertical [b] | |
| F1-1 | LLLF | 2.8–3.6 | 0.6–1.3 | 3.9 | 0.38 | He et al., 2000, 2010 |
| F1-2 | JQHF | 3.2–3.8 | 0.7–1.0 | 4.4 | / | He et al., 2000 |
| F1-3 | MMSF | 3.8–4.0 | / | 3.7 | / | He et al., 2000 |
| F1-4 | LHSF | 4.0–4.2 | / | 4.0 | / | Liu et al., 2018 |
| F1-5 | HYF | 3.6–4.2 | 0.2–1.0 | 3.2–4.5 | / | Li et al., 2009; Matrau et al., 2019 |
| F1-6 | LPSF | 2.0–3.6 | 0.2–1.1 | 0.7–3.0 | 0.2–0.9 | Wang, 2018; Wang et al., 2021; |
| F1-7 | GG-BJF | 0.6–0.8 | 0.5–0.6 | / | / | / |
| F2-1 | ZZSF | - (1.2–1.4) | 0.2–0.3 | / | / | / |
| F2-2 | HHF | - (1.6–2.6) | 0–0.7 | / | 0.04–0.24 | Lei et al., 2014 |
| F2-3 | LSF | - (2.6–3.0) | / | - 2.2 | / | Min et al., 2003 |
| F2-4 | YWSF | - (2.6–3.0) | / | / | / | / |
| F2-5 | XGSF | - (1.4–3.0) | 0–0.4 | / | / | / |
| F3-1 | DTH-LXF | 0.8–1.8 | 0.5–0.8 | / | / | / |
| F3-2 | WQLF | 1.0–3.0 | 0.1–0.9 | 2.5–2.9 | / | Chen et al., 2019 |
| F4-1 | EKLF | 3.2–3.8 | 0–0.5 | 4.9 | 0.25 | Li, 2009 |
| F4-2 | TZF | 2.2–3.2 | 0.3–0.7 | 1.4–3.2 | 0.1–0.3 | Ren et al., 2013 |
| F5 | TJSF | 0–0.8 | 0.1–0.3 | 0.77–0.96 | 0.1–0.2 | Li et al., 2019; X. Li et al., 2017; Zhang et al., 2015 |
| F6 | WHLSF | - (0.6–0.8) | 0–0.2 | - 0.28 | 0.11 | Lei, 2016 |
| F7 | NSSF | - (0.4–0.6) | 0–0.3 | - 0.35 | 0.10 | Lei, 2016 |
| F8 | EHLSF | 0.2–0.4 | 0.1–0.5 | / | 0.88 | Lei et al., 2016 |
| F9 | ZYGF | 0.2–0.4 | 0–0.2 | / | / | / |
| F10 | LHTF | / | 0–0.1 | / | 0.18 | Lei et al., 2011 |
| F11 | YCF | / | 0–0.2 | / | 0.14 | Lei et al., 2008 |
| F12 | QSHF | / | 0–0.2 | / | / | / |
| F13 | YTSF | / | 0–0.2 | / | / | / |
| F14-1 | ZLHF | - (0.6–0.8) | 0.2–0.6 | / | 0.12–0.51 | Hou et al., 1999 |
| F14-2 | MXSF | 0.2–0.4 | 0.1–0.4 | 0.5–1.72 | / | Song et al., 2006 |
| F15-1 | WLJSF | 0.4–0.6 | 0.4–0.5 | / | / | / |
| F15-2 | ELJSF | - (0.2–0.4) | 0.3–0.5 | / | / | / |
| F16-1 | DB-BLJF | 0.8–1.8 | 0.1–0.4 | 1.3 | 0.39 | Liu et al., 2015 |
| F16-2 | WD-KXF | 0–0.2 | 0-0.1 | 1.0 | / | Zheng et al., 2016 |

[a] Positive value indicates left-lateral slip rate. [b] Fault attributes are shown in Figure 11b




## 4 Discussion

### 4.1 Comparison with previous results

The boundary conditions of our model were derived from the GPS data of Wang et al. (2020). Figure 12 shows a comparison of the results of our model with the GPS data. The data are consistent both regarding the direction and magnitude. To examine the fit between the model results and GPS data more closely, we selected a SW–NE profile that covers the study area (Figure 12, C–C') and projected all GPS-observed values within 50 km onto the profile. Figure 13 shows that the simulated and observed values on either side of the profile (i.e., locations close to the model's

boundaries) are identical. There were differences between the simulated and GPS-observed values, but the differences were like the margin of error for the GPS data. Based on comparisons between the simulated and GPS-observed values in the planar map and cross-sectional profile, the kinematics simulated with our model agree with the GPS data.

Table 3 is a compilation of fault slip rates obtained using geologic methods. It shows that the simulated horizontal and vertical fault slip rates strongly correlate with the slip rates obtained by geologic approaches. For example, the

horizontal slip rates simulated for major faults along the F1 fault (e.g., the Laohushan, Haiyuan, and Liupanshan faults) are similar to the slip rates obtained by single-point measurements with geological methods (Table 3). The slip rates (2.6–3.0 mm/a) simulated for the F2-2 Luoshan Fault are similar to the measured slip rate (2.2 mm/a) and the slip rates simulated for the West Qinling Fault in the Zhangxian and Tianshui region (2.4–3.0 mm/a, Figure 10) are consistent with the slip rates obtained using geological methods (2.5–2.9 mm/a; Chen et al., 2019).

### 310 4.2 Fault slip rates and seismic hazards

#### 4.2.1 Tianzhu Seismic Gap

The M8.0 Gulang earthquake occurred in 1927 in the northwestern part of the F1 fault, whereas the M8.0 Haiyuan earthquake occurred in 1920 on the Haiyuan Fault. The Jinqianghe, Maomaoshan, and Laohushan faults, which are in the region between these earthquake zones, participated in neither, are collectively known as the Tianzhu Seismic Gap

(TSG, Guo et al., 2019; Y. Li et al., 2016; Figure 14a). Based on the model simulation, the left-lateral strike-slip rates of the Jinqianghe, Maomaoshan, and Laohushan faults were 3.2–3.8, 3.8–4.0, and 4.0–4.2 mm/a, respectively (Table 2). Therefore, all three faults have relatively high slip rates compared with the rest of the study area. Based on the slip rates and other fault data, we estimated the earthquake magnitude based on the energy accumulated during elapsed time (Purcaru et al., 1978) and recurrence intervals (Shen et al., 2009), as shown in Table 4. The Jinqianghe,

Maomaoshan, and Laohushan faults can generate $M_S6.9$, $M_S7.2$, and $M_S6.8$ earthquakes, with recurrence intervals of 707, 890, and 1132 years, respectively. These intervals are like the 1000-year recurrence interval estimated based on geological evidence (Liu-Zeng et al., 2007). It has been reported that 675 and 952 years have elapsed since the Jinqianghe and Maomaoshan faults were last activated (Gan et al., 2002; Table 4). Therefore, the likelihood these two faults will reactivate in the next few decades is high. For the Laohushan Fault, it is believed that the most recent

seismic event inducing surface ruptures was the 1888 M6.25 Jingtai earthquake. Because only 133 years have elapsed since this event, the near-term seismic hazard of this fault is low. However, several researchers believe the TSG experienced a M ≥ 8.0 earthquake with a recurrence interval of 1000 years, which would involve the simultaneous

activation of the Lenglinglong, Jinqianghe, Maomaoshan, and Laohushan faults (Chen, 2014). Based on historical

seismic records, 925 years have elapsed since the most recent M ≥ 8.0 simultaneous rupture event in the TSG (Liu et

al., 2018). Therefore, the likelihood that a major earthquake will occur in this region in the next few decades is high.

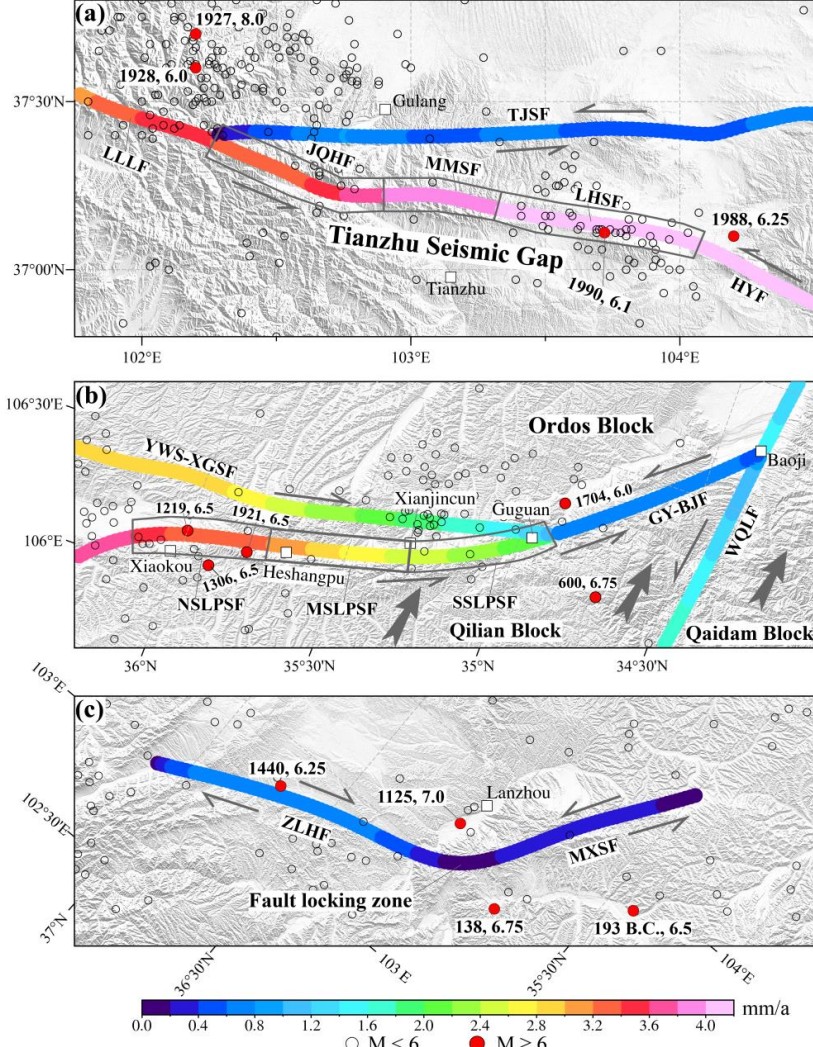

**Figure 14. (a) Distribution of fault slip rates and historical earthquakes in the Tianzhu Seismic Gap. (b) Distribution of fault slip rates and historical earthquakes at the Liupanshan, Guguan–Baoji, and Yunwushan–Xiaoguanshan faults. The gray arrows represent the movements of the Qilian and Qaidam blocks and NSLPSF, MSLPSF, and SSLPSF are the**
**northern, middle, and southern segments of the Liupanshan Fault, respectively. (c) The locked fault zone at the junction of the Maxianshan and Zhuanglanghe faults was formed by the left-parallel strike-slip motions of the former and right-parallel strike-slip motion of the latter.**

### 4.2.2 Seismic gap at the southern Liupanshan and Guguan–Baoji faults

Because of the unique tectonic setting of the Liupanshan and Guguan–Baoji faults, this region is simultaneously
affected by three sources of tectonic stress (Figure 14b). First, the SE movement of the Qilian Block horizontally





compresses this region against the stable Ordos Block to the east and causes strong thrusting activity (Figure 6a).
Second, the Liupanshan Fault zone at the SE end of the Haiyuan Fault and the displacements caused by the left-lateral
strike-slip faulting of the Haiyuan Fault at the SE end of its SW wall are accommodated by the shortening of the
Liupanshan Fault zone. Third, our simulations show that the Yunwushan–Xiaoguanshan Fault has a significant right-
lateral strike-slip component (Figure 7), which contributes to the accumulation of right-lateral shear strain in the
Liupanshan and Guguan–Baoji fault zones (Du et al., 2018). Based on these three sources of tectonic stress, the
Liupanshan and Guguan–Baoji faults are prime locations for stress accumulation. The distribution of the velocities of
the faults are also indicative of stress accumulation in this region (Figure 7). The northern segment of the Liupanshan
Fault has slip rates of 3.2–3.6 mm/a, which suddenly decrease to 2.5 mm/a in the middle and southern segments of
the fault. In addition, the slip rate of the Guguan–Baoji Fault is only 0.7 mm/a. The northern part of the Xiaoguanshan
Fault has slip rates of 2.8–3.0 mm/a, which decrease to 1.5 mm/a in the southern part (Figure 14b). These changes in
the slip rate indicate that the middle–southern segments of the Liupanshan and Guguan–Baoji faults have high slip
rate deficits, implying this region accumulates strain very rapidly. However, in terms of seismic activity, only the
northern part of the Liupanshan Fault has a history of major earthquakes including an M7 earthquake in 1219, M6.5
earthquake in 1306, and M6.5 earthquake in 1921 (Figure 14b). Earthquakes stronger than M6.0 have not been
recorded in the middle–southern parts of Liupanshan Fault. Their seismic activity mainly manifests as small and sparse
earthquakes and the most recent activation was recorded 500 years ago (Shi et al., 2014). Minor earthquakes related
to the Guguan–Baoji Fault are scarce and the only notable earthquakes that have occurred near this fault are an M6.0
earthquake in 1704 and the 600 AD Qinlong M6.75 earthquake (Shi et al., 2013). It has been estimated that 1400 years
have elapsed since its last activation (Xue, 2014).

**Table 4.** Earthquake magnitude and recurrence period of each fault based on the energy accumulated during
the elapsed time since the last earthquake

| Fault name | $V_1$ (mm/a) | $V_2$ (mm/a) | $L_1$ (km) | $L_2$ (km) | $\mu$ (Gpa) | t | S (m) | $M_S$ | T (a) |
|---|---|---|---|---|---|---|---|---|---|
| JQHF | 3.5 | 1.4 | 34 | 20 | 34.5 | 675 | 1.5 | 6.9 | 707 |
| MMSF | 3.9 | 1.4 | 51 | 20 | 34.5 | 952 | 2.2 | 7.2 | 890 |
| LHSF | 4.1 | 1.4 | 70 | 20 | 34.5 | 133 | 3.1 | 6.8 | 1132 |
| MSLPSF, SSLPSF | 2.5 | / | 80 | 20 | 34.5 | 500 | 3.5 | 7.2 | 1397 |
| GG-BJF | 0.7 | / | 70 | 20 | 34.5 | 1400 | 3.1 | 7.1 | 4365 |

$V_1$ is the simulated average slip rate of the fault; $V_2$ is the aseismic creep rate of the fault (Gan et al., 2002);
$L_1$ is the length of the fault (Xu et al., 2016); $L_2$ is the width of the fault, measured at the dominant focal depth
of the region (Li et al., 2020); $\mu$ is the shear modulus of the rocks (Aki et al., 2002); t is the time that has
elapsed since the most recent earthquake (Gan et al., 2002; Shi et al., 2013, 2014; Wang et al., 2001); S is the
largest maximum coseismic slip, calculated using the method of Gan et al. (2002); Ms is the earthquake
magnitude corresponding to the energy accumulated by the fault between recurrences (Purcaru et al., 1978); T
is the recurrence interval of the fault, where $T = S/(V_1-V_2)$ (Shen et al., 2009). The fault names are defined in
Figure 1 and Figure 14.

The southern segments of the Liupanshan and Guguan–Baoji faults are in prime locations for stress/strain
accumulation and constitute a seismic gap on a major strike-slip fault. Furthermore, this region has a history of strong
earthquakes. Hence, it is a hazard zone for strong earthquakes. Based on the fault slip rates obtained from our model
and fault data in the literature, we estimated that the energy accumulated by the middle–southern Liupanshan and





365     Guguan–Baoji faults is sufficient to generate M$_S$7.2 and M$_S$7.1 earthquakes, with recurrence intervals of 1400 and
4429 years, respectively (Table 4). Because the next event is not likely to occur for a long time, the middle–southern
Liupanshan and Guguan–Baoji faults are most likely in a state of stress accumulation. Therefore, the likelihood of a
major earthquake in the next few decades is low in this region.

### 4.2.3 Maxianshan–Zhanglanghe fault zone

The Maxianshan Fault in Lanzhou is a large Holocene reverse strike-slip fault. It is an important earthquake-
controlling fault that affects and constrains the seismicity of this region (Yuan et al., 2003). Different left-lateral strike-
slip rates have been reported for this fault—3.73 mm/a (Yuan et al., 2002), 0.5–1.72 mm/a (Song et al., 2006), and
0.93 mm/a (Liang et al., 2008). These discrepancies may be attributed to the loess that covers the extension of the
fault, which obscures the fault traces in many segments and makes it difficult to track its activity. The simulations
performed in this study indicate that the left-lateral strike-slip rates of the Maxianshan Fault range from 0.2 to 0.4
mm/a and that the vertical slip rates vary from 0.1 to 0.4 mm/a (Figure 14c, Table 3). Therefore, the left-lateral strike-
slip motions of the Maxianshan Fault may not be as intense as previously thought, but they have a relatively large
thrust component. The Zhanglanghe Fault is predominantly a right-lateral strike-slip fault with slip rates ranging from
0.6 to 0.8 mm/a (Figure 14c, Table 3). These slip rates are significantly greater than the left-lateral strike-slip rates of
the Maxianshan Fault. Note that the junction between the Maxianshan and Zhuanglanghe faults is a locked fault zone
with a slip rate of zero. Locked fault zones accumulate high concentrations of stress. An earthquake will occur when
this stress exceeds the ultimate strength of the rocks in this segment. Some have suggested that the 1125 Lanzhou
M7.0 earthquake occurred in such a tectonic setting (He et al., 1997; Figure 13c). Given that the recurrence interval
of this region is 2250–3590 years and the last event was only 896 years ago (Liang et al., 2008), the near-term risk of
a major earthquake in this region is low because the next event is not expected to occur for a long time. The locked
fault zone jointly controlled by the Maxianshan and Zhuanglanghe faults represents a tectonic setting conducive for
strong quakes and is currently in a state of stress accumulation.

### 4.3 Implication for deformation block models

The focus of studies on the deformation and uplift mechanisms of the TP is on two hotly debated end-member
paradigms: the "block deformation" paradigm (Meade, 2007; Tapponnier, 2001; Tapponnier et al., 1982; Thatcher,
2007) and "continuous deformation" paradigm (England et al., 1986, 1988; Royden et al., 1997; Zhang et al., 2004).
The continuous paradigm argues that the deformation of the TP encompasses the entire lithosphere and region below
and that the deformation observed on the surface is actually "diffuse" deformation that stretches from the fault zone
to the insides of the blocks. In contrast, the block paradigm argues all deformations occur near fault zones and that the
strain inside the blocks is insignificant. This implies that the deformations of the TP should display "aggregated"
characteristics. Several researchers have also argued that the deformation of the continental crust indicates
superpositions between multiple modes, with the dominant mode varying in different regions (Deng et al., 2018; Wang
et al., 2020). Although end-member kinematic models for the deformation of the TP cannot be tested by simply using





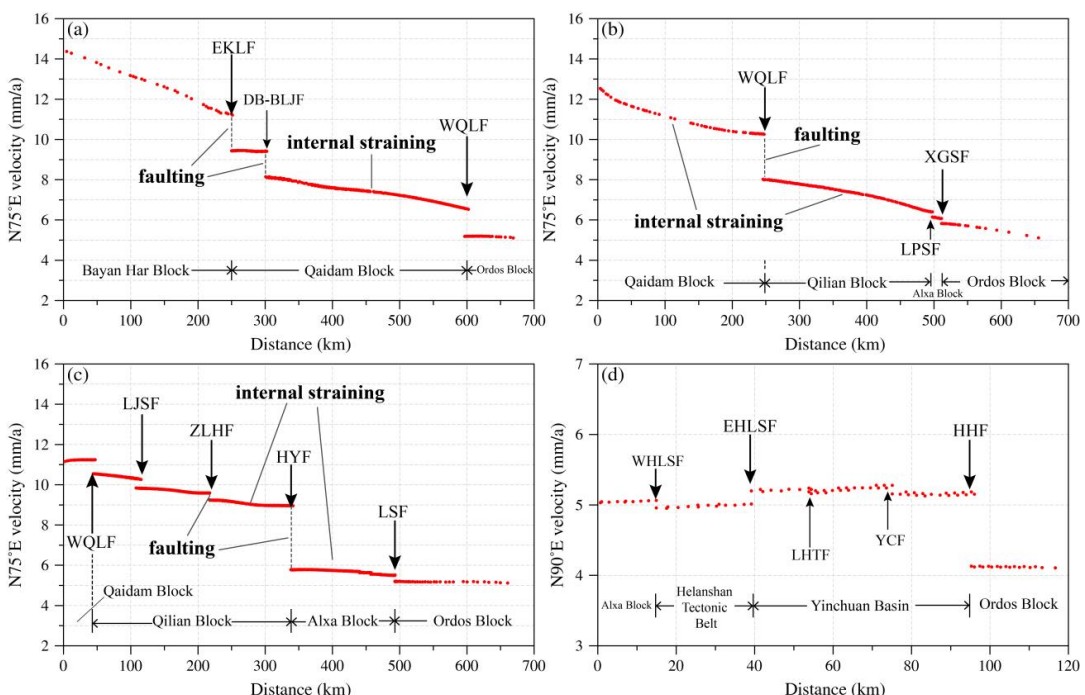

**Figure 15. Modeled velocity profiles across the study area with a N75°E orientation. The profiles in (a)–(d) correspond to the profiles AA', BB', CC', and DD' in Figure 12, respectively. The red dots indicate the N75°E component of the node motion velocity within 2 km on both sides of the profile. Fault names are defined in Figure 1.**

**Table 5.** Velocity differences generated by faults and internal deformations along each profile.

| | Total velocity difference (mm/a) | Velocity differences among faults (mm/a) | | Velocity differences inside blocks (mm/a) | |
|---|---|---|---|---|---|
| | | Value | Percentage | Value | Percentage |
| A-A' | 9.27 | 4.43 | 48% | 4.84 | 52% |
| B-B' | 7.46 | 2.77 | 37% | 4.69 | 63% |
| C-C' | 6.14 | 4.99 | 81% | 1.15 | 19% |

The location of each profile is shown in Figure 12.

absolute slip rates at block boundary faults (Yao et al., 2019), the crustal velocities and fault slip rates obtained from

geodynamics models can determine the ratio of internal block deformation to block boundary fault deformation.

In this study, we prepared four cross-sectional profiles in the N75°E direction that included the major faults and blocks of the study area and then projected the velocities of all nodes within 2 km of either side of these profiles onto the profile line. Figure 15 and Table 5 show the distribution of the velocity differences (i.e., crustal deformations) on the faults and inside the block for each of the profiles. All velocity differences on the profiles were exclusively derived

from fault slipping and internal block deformations. In the A–A' and B–B' profiles, the velocity differences caused by internal block deformations account for half of the total velocity differences. Therefore, the Bayan Har Block, Qaidam Block, and southeastern part of the Qilian Block accommodate most deformations caused by the SE





movement of the NETP through internal block deformations, which are "continuous deformations" (Figure 15a and 15b). In the C–C' profile, the velocity differences absorbed by fault slipping account for 81% of the total velocity differences and the velocity differences absorbed by slipping on the Haiyuan Fault alone account for 52% of the total velocity differences. This indicates that the crustal deformations of the Qilian Block, especially those near the Haiyuan Fault, are predominantly "block deformations" (Figure 15c). This result is consistent with those of previous studies that state that the deformations of the TP are continuous in inner regions and block-like in outer regions (Wang et al., 2020). The Haiyuan Fault could represent the northern plate boundary that extruded eastward from the TP. The D–D' profile shows that the tectonic deformations of the Helanshan–Yinchuan Basin structural belt slightly differ from those in other profiles. The NE expansion of the TP leads to near-N–S compression on the Helanshan–Yinchuan Basin structural belt (Yang, 2018), which causes the Yinchuan Basin to move eastwards faster than the Helanshan structural belt and Alxa Block. This manifests as an eastward extension in the Yinchuan Basin. The crustal deformations caused by this process are accommodated by the right-lateral strike-slip of Huanghe Fault (Figure 14d).

## 5 Conclusions

In this study, a detailed 3D geomechanical–numerical model of the NETP was constructed based on geophysical, geodetic, and geological data. This model accounts for physical rock properties, gravity fields, fault friction coefficients, initial crustal stresses, and boundary conditions. Special attention has been paid to the evaluation of fault friction coefficients and initial stress field parameters, which are important for kinematics simulations. To obtain the fault friction coefficients, simulations were performed using a series of friction coefficients. The results of these simulations were then compared to GPS observations. Friction coefficients with the lowest global fitting error were used in the final model. The initial stress field was characterized using the crustal model of Sheorey (1994) and the procedures of Hergert (2009). The initial stresses obtained based on this procedure agree well with the stress fields measured across the globe. Based on the numerical analysis of our model, we obtained the horizontal and vertical crustal velocities of the study area as well as the horizontal and vertical velocities of the major faults. The results were then validated against independent geodetic, geological, and paleomagnetic data.

Based on the analysis of the kinematics of the study area's major faults, the Laohushan and middle–southern Liupanshan faults as well as the Guguan–Baoji and locked fault zones at the junction of the Maxianshan and Zhuanglanghe faults are hazard zones for strong earthquakes. However, the likelihood of a major earthquake at these faults is low in the short term because they are currently in the stress accumulation state. In contrast, the Jinqianghe–Maomaoshan Fault will probably cause a ~M7.0 earthquake in the following decades. The simulation also provided information on the deformation modes of the NETP. Because of velocity differences between the opposing sides of the Haiyuan, West Qinling, and East Kunlun faults, as well as the relative stability of the Alxa and Ordos blocks, the NE expansion of the TP has caused the fault-separated Qilian, Qaidam, and Bayan Har blocks to extrude in the SEE direction and rotate in the clockwise direction. The crustal deformations at the NETP are predominantly continuous in the Bayan Har and Qaidam blocks and predominantly block-like in the Qilian Block.





**Data Availability**

The GPS data displayed in Figure 4 and Figure 12 are available through Wang and Shen (2020). The fault traces were obtained from Xu et al. (2016). The CRUST1.0 was obtained from Laske et al. (2013).

**Author contribution**

LL and XL contributed to the model building. LL carried out the analysis, wrote the paper, and prepared the figures. FY, LP and JT reviewed and edited the paper.

**Competing interests**

The authors have no competing interests to declare.

**Disclaimer**

Publisher's note: Copernicus Publications remains neutral with regard to jurisdictional claims in published maps and institutional affiliations.

**Acknowledgments**

This research was funded by the Natural Science Foundation of Ningxia Province (grant number: 2020AAC03445, 2021AAC05022). Some figures were plotted using The Generic Mapping Tools (https://www.generic-mapping-tools.org/). Slip rates on fault surfaces were calculated by the software GeoStress (Stromeyer et al., 2020).

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
