# Peer review of "Numerical Simulation of Present-day Kinematics at the Northeastern Margin of the Tibetan Plateau"

_Solid Earth, 2022_

## Referee Comment (RC1)

In the manuscript, Li et al constructed a 3-D numerical model to simulate the crustal deformation in northeastern Tibet. Their modeling results present fault slip rate along several large strike-slip faults and relatively regional crustal faults; based on the above, they quantitively assessed seismic hazards along several faults. In general, I think the topic and scientific issues the authors discussed are interesting. Nonetheless, based on my research, I think there are several major issues the authors should make clear.

1) Regarding the 3-D model. The main topic of the manuscript is to investigate the 'present-day' fault slip rates. However, in constructing the numerical model, the authors assume low fault friction coefficient, thus allowing faults slip aseismically or continuously in the seismogenic layer. Such a practice is inconsistent with what we usually think of as interseismic fault deformation, because during the interseismic phase, faults are locked in the seismogenic zone and freely slipping below it. However, in this study, the fault slip rates are due to average velocities over several seismic cycles. Therefore, the authors should consider their model either reflects long-term kinematics or update the model by locking the faults in the seismogenic layer.

2) A relevant issue that promotes me to judge the model assumption is the GPS velocity profiles shown in figure 13 and 15. The modeled velocities behave as steps across faults, and therefore show discrepancies with GPS observations. The velocity steps are related to the model, which allows faults slip continuously in response to the far-field loading they experience.

3) A subsequent issue based on the modeling results is the seismic hazards assessment. If the faults are slip continuously in the seismogenic zone, how does elastic strain accumulate? I gauss the authors might mis-interpreted the fault slip rate and fault locking; because in calculating strain budget on the Jinqianghe-Maomaoshan-Laohushan faults, they used the 3.5-4.1 mm/a long-term slip rate as stress loading rate (actually in their model, the faults are freely slipping in the seismogenic layer); whereas in interpreting the seismic potential on the Maxianshan-Zhuanglanghe faults, they regarded zero slip rate as reflective of locked fault zone. The above practices are contradictory.

4) Another major issue is the block deformation. In the abstract, the author state that the Bayan Har and Qaidam blocks are deforming continuously, whereas the Qilian block is more of block-like. The main evidences for this conclusion come from the interpretation of velocity gradient within blocks (section 4.3). I disagree with such interpretations, because crustal blocks are rotating with reference to their Euler poles, the velocity gradient within blocks are likely caused by the block rotations. Therefore, unless the authors separate the block rotational components, the velocity gradient inside the block is misinterpreted.

5) There are also quite a lot of language and/or grammar issues. I would suggest the authors seek help from native speakers or professional services.
6) Based on my above judgements, I suggest a major revision for the manuscript.

There are also lots of language and/or grammar and other minor issues. I just name a few:
1) Line27-28, give references
2) Line34, please explicitly indicate the earthquake locations in figure 1
3) Line36, what is the strength of an earthquake?
4) Line39, the sentence reads quite strange, rephase
5) Line48-49, actually, quite a lot previous studies adopted elastic block models (e.g., Y. Li et al., 2017, 2021), and their results show non-negligible internal deformation
6) Line58, partitioning of deformation modes? What does it mean?
7) Figure1, indicate the time span of earthquakes and the sources, give descriptions of P1-P3
8) Line84-85, cite references
9) Table1, list the references in the table
10) Line125, change critically important to important
11) Line134, fitting misfit in mm/a or cm/a or others?
12) Line135, I am not fully understood, why F2 and F3 are not considered in friction coefficients adjustments?
13) Line136, it seems to me, for F3, friction coefficient from 0.02 to 0.1 is large, why?
14) Line163, Wang et al. (2020) should be Wang and Shen (2020), check the whole manuscript to avoid similar mistakes
15) Line180, see the 5$^{th}$ major comment. I don't agree with this interpretation, velocity gradient within blocks might be related to block rotation as well!
16) Line187, crustal velocity, not crust speed. Check and replace the whole manuscript
17) Line215, older?
18) Line215-217, give reasons for the discrepancies
19) Table3, change table to figure, which shows 1:1 plots of fault slip rates
20) Line297, rephase the sentence
21) Line300, see the 1$^{st}$ and 2$^{nd}$ major comments. It seems to me that the differences are large, especially across faults. The modeled velocities have steps across faults, this should be result from the fact that numerical model does not consider fault locking.
22) Line315-319, I don't think the way of earthquake potential assessment appropriate. First, aseismic creep is not found along the Jinqianghe and Maomaoshan faults, as recent studies show. Second, the seismogenic does not corresponds to 20 km. Check the latest studies (e.g., Y. Li, 2021, JGR) to update your way of calculation.

The above modifications are not sufficient to represent expressive or language problems

in the text, please the author see the revision as an opportunity to carefully re-arrange your language and contents.

Yanchuan Li

---

## Referee Comment (RC2)

Review for se-2022-17

The article titled "Numerical Simulation of Present-day Kinematics at the Northeastern Margin of the Tibetan Plateau" focuses on the slip rates of active faults at the northeastern margin of the Tibetan Plateau. It is very important to understand the lateral expansion of the Tibetan Plateau and assess the seismic hazards in this region. By use of a three-dimensional geomechanics-numerical model, the authors obtained the horizontal and vertical crustal velocities and slip rates of active faults in the study area. The results are closely consistent with the observation in the area. The reviewer considers that this article is worth publishing on Solid Earth.

The authors collected and researched previous relative works in this area and constructed a reasonable numerical model. The design for the geometric layering and block division, the rock properties, the boundary condition and other parameters are suitable. The results are reasonable and important for scientific research and seismic hazard analysis in this area.

It is a pity that the authors have not paid attention to the important information given in Fig.11 during they discuss the "Fault slip rates and seismic hazards" in 4.2. Theoretically research and practical observations show that the isolated uplift area is the most dangerous place for earthquakes. So the review suggests that the authors pay attention to the result of the vertical velocity and slip rate of active faults in Fig. 11. The intersection location of areas with positive velocity value and the relative active fault is the most hazardous seismic location. The reviewer strongly suggests the authors reevaluate the earthquake risk regions by consideration of this factor combined with others. Another suggestion is that the authors can illustrate what finite element software is used in the modeling and the reason.

Generally speaking, this is a very good article both in the modeling and in the research area.

---

## Editor Comment (EC1)

[revised manuscript text omitted]

---

## Author Comment (AC1)

Dear Reviewer (se-2022-17),

We appreciate the time and effort you have spent in reviewing this manuscript. The comments and suggestions helped us improve the manuscript significantly. Our point-by-point responses to the comments are shown below.

**1. Reviewer's comment:** Regarding the 3-D model. The main topic of the manuscript is to investigate the 'present-day' fault slip rates. However, in constructing the numerical model, the authors assume low fault friction coefficient, thus allowing faults slip aseismically or continuously in the seismogenic layer. Such a practice is inconsistent with what we usually think of as interseismic fault deformation, because during the interseismic phase, faults are locked in the seismogenic zone and freely slipping below it. However, in this study, the fault slip rates are due to average velocities over several seismic cycles. Therefore, the authors should consider their model either reflects long-term kinematics or update the model by locking the faults in the seismogenic layer.

**Authors' reply:**

Thanks for pointing this out.

The modeled velocities in this paper refer to long-term velocities over several seismic cycles. We mistakenly used the word "present-day" in the title of the original manuscript and we have changed it to "contemporary".

**2. Reviewer's comment:** A relevant issue that promotes me to judge the model assumption is the GPS velocity profiles shown in figure 13 and 15. The modeled velocities behave as steps across faults, and therefore show discrepancies with GPS observations. The velocity steps are related to the model, which allows faults slip continuously in response to the far-field loading they experience.

**Authors' reply:**

The locking of the seismogenic zone of the fault during the interseismic phase gives rise to elastic strain accumulation effects that cause across-fault velocity gradients to be smooth. However, in this paper, the faults were set with friction coefficients and allowed to slip to simulate long-term slip rates over several seismic cycles. That is, some of the elastic strain that accumulated on the fault during the interseismic phases is released in the form of fault slip. Therefore, it is reasonable for the modeled velocities to have steps across faults (Thatcher et al., 1999; Hergert and Heidbach, 2010).

Many practices have also indicated that the approach used in the model is feasible (Hergert and Heidbach, 2010; Hergert et al., 2011; Li, Hergert, et al., 2021).

**3. Reviewer's comment:** A subsequent issue based on the modeling results is the seismic hazards assessment. If the faults are slip continuously in the seismogenic zone, how does elastic strain accumulate? I guess the authors might mis-interpreted the fault slip rate and fault locking; because in calculating strain budget on the Jinqianghe-Maomaoshan-Laohushan faults, they used the 3.5-4.1 mm/a long-term slip rate as stress loading rate (actually in their model, the faults are freely slipping in the seismogenic layer); whereas in interpreting the seismic potential on the Maxianshan-Zhuanglanghe faults, they regarded zero slip rate as reflective of locked fault zone. The above practices are contradictory.

**Authors' reply:**

We believe that whether the seismogenic zone of a fault is locked depends on the time scale we set. It is obviously that the seismogenic zone is locked during the interseismic period observed by GPS. However, the seismogenic zone can also be considered as unlocked on a long-term time scale, especially when the coseismic displacements of multiple earthquakes can completely compensate for the slip deficit accumulated during fault locking periods (Wang, 2021). Actually, the conclusion that large faults have extremely low effective friction coefficients have been recognized by more and more studies over the past two decades (He et al., 2013; Li et al., 2015; Wang, 2021).

In this paper, the modeled velocity of the faults refers to long-term slip rate which means the sum of the interseismic velocity and coseismic displacement over a period of several seismic cycles. The seismogenic zone of the faults are locked during inderseismic period, allowing elastic strain to accumulate.

Regarding the "Locked fault zone" on the Maxianshan-Zhuanglanghe faults, we made an inappropriate description due to language problems. Lines 379–383 of the original manuscript have been rephrased as follows.

*Note that the slip rate on the junction between the Maxianshan fault and Zhuanglanghe fault is almost zero. It can be inferred that the junction area would accumulate high concentrations of stress under the continuous eastward movement of the Qilian Block. An earthquake will occur when this stress exceeds the strength of the rocks in this segment. Some have suggested that the 1125 Lanzhou M7.0 earthquake occurred in such a tectonic setting (He et al., 1997; Fig. 14c).*

**4. Reviewer's comment:** Another major issue is the block deformation. In the abstract, the author state that the Bayan Har and Qaidam blocks are deforming continuously, whereas the Qilian block is more of block-like. The main evidences for this conclusion come from the interpretation of velocity gradient within blocks (section 4.3). I disagree with such interpretations, because crustal blocks are rotating with reference to their Euler poles, the velocity gradient within blocks are likely caused by the block rotations. Therefore, unless the authors separate the block rotational components, the velocity gradient inside the block is misinterpreted.

**Authors' reply:**

Thanks for pointing this out. We did ignore the effect of block rotation in the original manuscript, and the Section 4.3 has been revised as follows.

[Figure]

**Figure 15. Modeled velocity profiles across the study area with orientation of profiles. The profiles in (a)–(d) correspond to the AA', BB', CC', and DD' in Fig. 12, respectively. The red dots indicate the components along the profiles of the node motion velocity within 2 km on both sides of the profile. The green dots represent the velocity component along the profiles due to plate rotation. The blue dots indicate the differences between the red and green dots. Fault names are defined in Fig. 1.**

*4.3 Implication for deformation mechanism of NETP*

*The deformation of NETP is the result of the combined action of block rotation, faulting, and the intrablock straining (Meade and Loveless, 2009). We analyzed four velocity profiles to compare the contributions of block rotation, faulting, and intrablock straining to the total deformation of NETP (Fig. 15). It is noted that the rigid displacements caused by block rotation were calculated according to the Euler pole locations and rotation rates with respect to the Eurasia plate (Wang et al., 2017; Y. Li et al., 2022), as shown in Fig. 15 a–d. The velocity gradient caused by block rotation accounts for more than 80% of that on the profiles. Obviously, the block rotation should*

*be the primary mechanism for the deformation of the NETP, which is similar to the southeastern Tibet (Z. Zhang et al., 2013). However, the intrablock straining of Bayan Har and Qaidam blocks contribute approximately 4 mm/a and 3 mm/a shortening in profiles of AA', BB' (Fig. 15a–b). The Qilian block also has a contribution of 2 mm/a shortening in profile BB' but decreases to about 1mm/a in profile CC' (Fig. 15b–c). Therefore, the intrablock straining is still significant for regional deformation. The boundary faults of the blocks, such as the East Kunlun fault, Haiyuan fault, West Qinling fault, also play an important role in regulating the deformation differences between blocks.*

*The D–D' profile shows that the tectonic deformations of the Yinchuan Basin structural belt slightly differ from those in other profiles. The NE expansion of the TP leads to near-N–S compression on the Yinchuan Basin (Yang, 2018), which causes it to move eastwards faster than the Alxa Block. This manifests as an eastward extension in the Yinchuan Basin. The crustal deformations caused by this process are accommodated by the right-lateral strike-slip of Huanghe Fault (Fig. 15d).*

**5. Reviewer's comment:** There are also quite a lot of language and/or grammar issues. I would suggest the authors seek help from native speakers or professional services.
**Authors' reply:**
Thanks for your suggestion, we have used the services of a professional English editing company to improve the language of the manuscript.

**6. Reviewer's comment:** Based on my above judgements, I suggest a major revision for the manuscript.
**Authors' reply:**
Thanks for your valuable comments and suggestions.
We have carefully revised the full text, please pay attention to the revised manuscript.

**7. Reviewer's comment:** There are also lots of language and/or grammar and other minor issues. I just name a few:
**7.1 Reviewer's comment:** Line27-28, give references
**Authors' reply:**
Thanks for your suggestions. We have reorganized the language there and added references as follows.

*Having experienced the strong Cenozoic deformation, crust of this area develops a complex fault system with several large and deep faults, such as the generalized Haiyuan fault (F1), West Ordos fault (F2), West Qinling fault (F3), East Kunlun fault (F4), that divide the NETP into the Alxa, Ordos, Qilian, Qaidam, and Bayan Har blocks (Zhang et al., 2003; Fig.1). These faults are characterized by extremely intense tectonic movements and seismic activities (Zhang, 1999; Zheng et al., 2016b).*

**7.2 Reviewer's comment:** Line34, please explicitly indicate the earthquake locations in figure 1.

**Authors' reply:**
We have labeled all earthquakes greater than magnitude 8 on Fig. 1 as follows.

[Figure]

Figure 1. Earthquakes with magnitude M≥ 8.0 are labeled.

**7.3 Reviewer's comment:** Line36, what is the strength of an earthquake?
**Authors' reply:**
We have rephrased the sentence as follows.

*Since the generation and magnitude of an earthquake is closely related to fault activity, long-term fault slip rate plays a key role in medium- and long-term seismic hazard assessment (Ding et al., 1993; Xu et al., 2018).*

**7.4 Reviewer's comment:** Line39, the sentence reads quite strange, rephrase
**Authors' reply:**
We have rephrased the sentence as follows.

*For example, combined with coseismic displacements, long-term fault slip rates can be used to calculate earthquake recurrence interval (Shen et al., 2009) and assess the magnitudes of potential earthquakes (Bai et al., 2018; Hergert and Heidbach, 2010).*

**7.5 Reviewer's comment:** Line48-49, actually, quite a lot previous studies adopted elastic block models (e.g., Y. Li et al., 2017, 2021), and their results show non-negligible internal deformation
**Authors' reply:**
Thanks for pointing this out. We have rephrased the sentences as follows.

*For example, the geological slip rates only represent the activities of one fault branch that measured in a fault zone, which is always consist of several branches. They are usually lower than the geodetic slip rates on the fault as a whole if a rigid block assumption is adopted in the geodetic inversion process (Shen et al., 2009). However, several crustal deformation studies conducted in TP demonstrated that the internal block deformation in the NETP cannot be ignored (Royden et al., 1997; Zhang et al., 2004; Y. Li et al., 2017, 2021).*

**7.6 Reviewer's comment:** Line58, partitioning of deformation modes? What does it mean?
**Authors' reply:**
In the original manuscript, we intended to analyze whether the deformation pattern of the NETP is a block model or a continuum model according to the distribution characteristics of the velocity gradient. In subsequent manuscript, we will make adjustments to this description as follows.

*Based on these results, we summarized the long-term crustal deformation characteristics in the NETP.*

**7.7 Reviewer's comment:** Figure1, indicate the time span of earthquakes and the sources, give descriptions of P1-P3.
**Authors' reply:**
Thanks for your suggestions. We updated the Figure 1 caption as follows:

*Figure 1. Map of active faults and earthquakes of the NETP. Black lines represent the active faults. The light blue, red dots and the white pentagrams represent earthquakes from 1831 BC to 2017 AD from the National Earthquake Data Center (http://data.earthquake.cn). Black crosses (P1-P4) indicate the locations of four test sites for the comparison with the numerical model shown in Figure 3b. Faults discussed in the text are labeled as followed……*

**7.8 Reviewer's comment:** Line84-85, cite references

**Authors' reply:**

We will add corresponding references. The updated text is shown below.

*Lithospheric faults (i.e., F1, F2, F3, and F4) cut through the Moho and reach the bottom of the model (Zhan et al., 2005; B. Liu et al., 2017; Zhao et al., 2015; Fig. 2a, b; Table 1). All other faults are crustal faults that terminate in the upper, middle, or lower crust (Yuan et al., 2002b, 2003; Lease et al., 2012; Meng et al., 2012; B. Liu et al., 2017; Wu et al., 2020).*

**7.9 Reviewer's comment:** Table1, list the references in the table

**Authors' reply:**

We added a column to the right of the table for references.

**Table 1.** Geometric parameters of faults in the model

|  | Fault name | Strike | Dip direction | Dip (°) | Reference |
|---|---|---|---|---|---|
| F1 | / | NWW-SSE | SW | 70 | RGAFSAO, 1988 |
| F2-1 | ZZSF | N-S | W | 70 | Gao, 2020 |
| F2-2 | HHF | N-S | W | 70 | Bao et al., 2019 |
| F2-3 | LSF | N-S | W | 80 | Wang et al., 2013 |
| F2-4 | YWSF | N-S | W | 70 | NIGS, 2017 |
| F2-5 | XGSF | N-S | W | 70 | NIGS, 2017 |
| F3-1 | DTH-LXF |  |  |  | Zhou et al., 2009 |
| F3-2 | WQLF | NWW-SSE | NE | 70 | Li, 2005 |
| F4-1 | EKLF | NW-SE | NE | 75 | Z. Liu et al., 2017 |
| F4-2 | TZF |  |  |  | J. Li et al., 2019 |
| F5 | East TJSF | NW-SE | SW | 70 | RGAFSAO, 1988 |
|  | West TJSF | E-W | S | 70 | RGAFSAO, 1988 |
| F6 | WHLSF | N-S | W | 80 | Lei, 2015 |
| F7 | NSSF | NW-SE | SW | 70 | RGAFSAO, 1988 |
| F8 | EHLSF | NE-SW | SE | 60 | Du, 2010 |
| F9 | ZYGF | E-W | S | 60 | NIGS, 2017 |
| F10 | LHTF | NNE-SSW | SE | 70 | NIGS, 2017 |
| F11 | YCF | NNE-SSW | NW | 70 | NIGS, 2017 |
| F12 | YTSF | NW-SE | SW | 65 | NIGS, 2017 |
| F13 | QSHF | NW-SE | SW | 45 | Tian et al., 2020 |
| F14-1 | ZLHF | NNW-SSE | SW | 45 | Xu et al., 2016 |
| F14-2 | MXSF | NW-SE | SW | 80 | Hou et al., 1999 |
| F15 | LJSF | NWW-SSE | SW | 50 | Yuan et al., 2005 |
| F16-1 | DB-BLJF | NW-SE | SW | 70 | Yuan et al., 2007 |
| F16-2 | WD-KXF | E-W | SW | 70 | Jia et al., 2012 |

The detailed fault names are defined in Fig. 1.

**7.10 Reviewer's comment:** Line125, change critically important to important
**Authors' reply:**
Thanks for your suggestion. We have changed it.

**7.11 Reviewer's comment:** Line134, fitting misfit in mm/a or cm/a or others?
**Authors' reply:**
The *misfit* is a dimensionless unit. It was calculated as follows (Cianetti et al., 2001):

$$misfit = \frac{\sum_i |\vec{V}_{GPS} - \vec{V}_{mod}|}{\sum_i |\vec{V}_{GPS}| + \sum_i |\vec{V}_{mod}|}$$

**7.12 Reviewer's comment:** Line135, I am not fully understood, why F2 and F3 are not considered in friction coefficients adjustments?
**Authors' reply:**
I guess you probably meant to refer to F2 and F4?

Actually, F1–F4 were all considered in the numerical simulation tests of friction coefficients adjustments. However, the adjustment of friction coefficients of F2 and F4 did little help to reduce the misfit value. Therefore, the friction coefficients of F2 and F4 remained unchanged.

**7.13 Reviewer's comment:** Line136, it seems to me, for F3, friction coefficient from 0.02 to 0.1 is large, why?
**Authors' reply:**
Let's first correct a mistake in the *misfit* calculation in Figure 3a of the original manuscript. Due to a coding error, the calculation of *misfit* in the original manuscript only considers the easting components of the GPS observations and modeled velocities. Now we have corrected the error and the updated Fig. 3a is shown in Fig. R2. The value of 0.02 can still be considered as the best friction coefficient.

[Figure]

Figure 3. A friction coefficient of 0.02 or 0.03 for all faults yields the smallest fitting error.

Now let's answer your question about "Line136, it seems to me, for F3, friction coefficient from 0.02 to 0.1 is large, why".

We determined this value through multiple numerical simulation tests. We found that increasing the friction coefficient of F3 and decreasing the friction coefficient of F1 is beneficial to the reduction of misfit. The simulation tests are as follows (Table R1). The misfit value is the lowest when the friction coefficients of F1 and F3 are 0.01 and 0.1, respectively.

Table R1. The simulation tests to find the lowest misfit

| $\mu'$ | F3=0.04 | F3=0.05 | F3=0.06 | F3=0.07 |
|---|---|---|---|---|
| *misfit* | 0.6078 | 0.05965 | 0.05855 | 0.05749 |
| $\mu'$ | F3=0.07; F1=0.01 | F3=0.08; F1=0.01 | F3=0.09; F1=0.01 | F3=0.10; F1=0.01 |
| *misfit* | 0.05639 | 0.05542 | 0.05449 | 0.05362 |

**7.14 Reviewer's comment:** Line163, Wang et al. (2020) should be Wang and Shen (2020), check the whole manuscript to avoid similar mistakes.
**Authors' reply:**

Thanks for pointing this out. We have corrected all similar mistakes.

**7.15 Reviewer's comment:** Line180, see the 5th major comment. I don't agree with this interpretation, velocity gradient within blocks might be related to block rotation as well!
**Authors' reply:**

Thanks for pointing this out. We have removed the relevant text.

**7.16 Reviewer's comment:** Line187, crustal velocity, not crust speed. Check and replace the whole manuscript
**Authors' reply:**

Thanks for pointing this out. We have checked the whole manuscript and made changes.

**7.17 Reviewer's comment:** Line215, older?
**Authors' reply:**

Thanks for pointing this out. We have replaced it with "earlier".

**7.18 Reviewer's comment:** Line215-217, give reasons for the discrepancies
**Authors' reply:**

We believe that there are two reasons for this.

First, early studies were based on geological methods with larger time scale. Second, the slip rate of the fault could not be better constrained in the past due to limited data (Li, et al., 2009).

We have already added it to the updated manuscript.

**7.19 Reviewer's comment:** Table3, change table to figure, which shows 1:1 plots of fault slip rates

**Authors' reply:**

Thanks for your suggestion.

We have carefully considered this issue, referring to the Figure 3 in the paper of Y. Li et al. (2021). The difference between the results obtained by the two methods can be clearly intuitively observed from the figure.

However, the modeled slip rates vary with the locations of the faults in this paper. It is also inappropriate to use a mean value to replace the slip rate of the entire fault. Therefore, we retained the Table3 used in the original manuscript.

**7.20 Reviewer's comment:** Line297, rephase the sentence

**Authors' reply:**

Thanks for your suggestion. We have rephased the sentence as follows.

*In order to further examine the fit between the model results and GPS data, we selected a NE–SW profile that crosses through the study area (Fig. 12, C–C') and projected all GPS-observed values within 50 km of both sides of the profile.*

**7.21 Reviewer's comment:** Line300, see the 1st and 2nd major comments. It seems to me that the differences are large, especially across faults. The modeled velocities have steps across faults, this should be result from the fact that numerical model does not consider fault locking

**Authors' reply:**

It has been explained before.

Please see the replies to comment 2 and 3.

**7.22 Reviewer's comment:** Line315-319, I don't think the way of earthquake potential assessment appropriate. First, aseismic creep is not found along the Jinqianghe and Maomaoshan faults, as recent studies show. Second, the seismogenic does not corresponds to 20 km. Check the latest studies (e.g., Y. Li, 2021, JGR) to update your way of calculation.

**Authors' reply:**

Thanks for pointing this out.

The aseismic creep rates have been updated according to the latest study (Y. Li, et al., 2021). We also learned that the locking depth of the Laohushan fault or the Tianzhu Seismic Gap is about 20–22 km according to Y. Li et al. (2017, 2021), which is approximately equal to the data used in our manuscript. The new calculation results are shown in the table below.

[revised manuscript text omitted]

---

## Author Response (AR1)

Journal: Solid Earth

Manuscript Number: SE-2022-17

Dear Editor and Reviewers,

We appreciate the time and effort you have spent in reviewing this manuscript. The reviewers' comments and suggestions helped us improve the manuscript significantly. We are pleased to inform you that we have considered all the comments and suggestions and revised the manuscript accordingly. It is worth mentioning that we have used the services of a professional English editing company to improve the language of the manuscript.

Please find our point-by-point response to the comments below. We hope that the revised manuscript is now ready for publication in *Solid Earth*.

Thank you for your consideration.

Yours sincerely,
Liming Li
School of Earth Sciences and Resources,
China University of Geosciences (Beijing),
No.29, Xueyuan Road, Haidian District, Beijing, P.R. China
Email: cugliming@cug.edu.cn

**Response to Comments from Reviewer #1**

Note that Section 3.2.4 of the original manuscript has been removed in this revision because we think that the F4 fault is not the primary research goal. Therefore, the sequence numbers of some figures may be changed in the revised manuscript.

**1. Reviewer's comment:** Regarding the 3-D model. The main topic of the manuscript is to investigate the 'present-day' fault slip rates. However, in constructing the numerical model, the authors assume low fault friction coefficient, thus allowing faults slip aseismically or continuously in the seismogenic layer. Such a practice is inconsistent with what we usually think of as interseismic fault deformation, because during the interseismic phase, faults are locked in the seismogenic zone and freely slipping below it. However, in this study, the fault slip rates are due to average velocities over several seismic cycles. Therefore, the authors should consider their model either reflects long-term kinematics or update the model by locking the faults in the seismogenic layer.

**Author's response and changes in manuscript:**

Thanks for pointing this out.
The modeled velocities in this paper refer to long-term velocities over several seismic cycles. We mistakenly used the word "present-day" in the title of the original manuscript and we have changed it to "contemporary" in the revised manuscript.

**2. Reviewer's comment:** A relevant issue that promotes me to judge the model assumption is the GPS velocity profiles shown in figure 13 and 15. The modeled velocities behave as steps across faults, and therefore show discrepancies with GPS observations. The velocity steps are related to the model, which allows faults slip continuously in response to the far-field loading they experience.

**Author's response:**

The locking of the seismogenic zone of the fault during the interseismic phase gives rise to elastic strain accumulation effects that cause across-fault velocity gradients to be smooth. However, in this paper, the faults were set with friction coefficients and allowed to slip to simulate long-term slip rates over several seismic cycles. That is, some of the elastic strain that accumulated on the fault during the interseismic phases is released in the form of fault slip. Therefore, it is reasonable for the modeled velocities to have steps across faults (Thatcher et al., 1999; Hergert and Heidbach, 2010).

Many practices have also indicated that the approach used in the model is feasible (Hergert and Heidbach, 2010; Hergert et al., 2011; Li, Hergert, et al., 2021).

**3. Reviewer's comment:** A subsequent issue based on the modeling results is the seismic hazards assessment. If the faults are slip continuously in the seismogenic zone, how does elastic strain accumulate? I guess the authors might mis-interpreted the fault slip rate and fault locking; because in calculating strain budget on the Jinqianghe-Maomaoshan-Laohushan faults, they used the 3.5-4.1 mm/a long-term slip rate as stress loading rate (actually in their model, the faults are freely slipping in the seismogenic layer); whereas in interpreting the seismic potential on the Maxianshan-Zhuanglanghe faults, they regarded zero slip rate as reflective of locked fault zone. The above practices are contradictory.

**Author's response:**

We believe that whether the seismogenic zone of a fault is considered locked depends on the time scale we set. It is obviously that the seismogenic zone is locked during the interseismic period observed by GPS. However, the seismogenic zone can also be considered as unlocked on a long-term time scale, especially when the coseismic displacements of multiple earthquakes can completely compensate for the slip deficit accumulated during fault locking periods (Wang, 2021). Actually, the conclusion that large faults have extremely low effective friction coefficients have been recognized by more and more studies over the past two decades (He et al., 2013; Li et al., 2015; Wang, 2021).

In this paper, the modeled velocity of the faults refers to long-term slip rate which means the sum of the interseismic velocity and coseismic displacement over a period of several seismic cycles. The seismogenic zone of the faults are locked during inderseismic period, allowing elastic strain to accumulate.

**Author's changes in manuscript:**

Regarding the "Locked fault zone" on the Maxianshan-Zhuanglanghe faults, we made an inappropriate description due to language problems. Lines 379–383 of the original manuscript have been rephrased as follows.

*Note that the slip rate on the junction between the Maxianshan fault and Zhuanglanghe fault is almost zero. It can be inferred that the junction area would accumulate high concentrations of stress under the continuous eastward movement of the Qilian Block. An earthquake will occur when this stress exceeds the strength of the rocks in this segment. Some have suggested that the 1125 Lanzhou M7.0 earthquake occurred in such a tectonic setting (He et al., 1997; Fig. 13c).*

**4. Reviewer's comment:** Another major issue is the block deformation. In the abstract, the author state that the Bayan Har and Qaidam blocks are deforming continuously, whereas the Qilian block is more of block-like. The main evidences for this conclusion come from the interpretation of velocity gradient within blocks (section 4.3). I disagree with such interpretations, because crustal blocks are rotating with reference to their Euler poles, the velocity gradient within blocks are likely caused by the block rotations. Therefore, unless the authors separate the block rotational components, the velocity gradient inside the block is misinterpreted.

**Author's response and changes in manuscript:**

Thanks for pointing this out. We did ignore the effect of block rotation in the original manuscript. The Section 4.3 has been revised as follows.

*4.3 Implication for deformation mechanism of NETP*
*The deformation of NETP is the result of the combined action of block rotation, faulting, and the intrablock straining (Meade and Loveless, 2009). We analyzed four velocity profiles to compare the contributions of block rotation, faulting, and intrablock straining to the total deformation of NETP (Fig. 14). It is noted that the rigid displacements caused by block rotation were calculated according to the Euler pole locations and rotation rates with respect to the Eurasia plate (Wang et al., 2017; Y. Li et al., 2022), as shown in Fig. 14 a–d. The velocity gradient caused by block rotation accounts for more than 80% of that on the profiles. Obviously, the block rotation should be the primary mechanism for the deformation of the NETP, which is similar to the southeastern Tibet (Z. Zhang et al., 2013). However, the intrablock straining of Bayan Har and Qaidam blocks contribute approximately 4 mm/a and 3 mm/a shortening in profiles of AA', BB' (Fig. 14a–b). The Qilian block also has a contribution of 2 mm/a shortening in profile BB' but decreases to about 1mm/a in profile CC' (Fig. 14b–c). Therefore, the intrablock straining is still significant for regional deformation. The boundary faults of the blocks, such as the East Kunlun fault, Haiyuan fault, West Qinling fault, also play an important role in regulating the deformation differences between blocks.*

*The D–D' profile shows that the tectonic deformations of the Yinchuan Basin structural belt slightly differ from those in other profiles. The NE expansion of the TP leads to near-N–S compression on the Yinchuan Basin (Yang, 2018), which causes it to move eastwards faster than the Alxa Block. This manifests as an eastward extension in the Yinchuan Basin. The crustal deformations caused by this process are accommodated by the right-lateral strike-slip of Huanghe Fault (Fig. 14d).*

[Figure]

**Figure 14. Modeled velocity profiles across the study area with orientation of profiles. The profiles in (a)–(d) correspond to the AA', BB', CC', and DD' in Fig. 11, respectively. The red dots indicate the components along the profiles of the node motion velocity within 2 km on both sides of the profile. The green dots represent the velocity component along the profiles due to plate rotation. The blue dots indicate the differences between the red and green dots. Fault names are defined in Fig. 1.**

**5. Reviewer's comment:** There are also quite a lot of language and/or grammar issues. I would suggest the authors seek help from native speakers or professional services.

**Author's response:**

Thanks for your suggestion, we have used the services of a professional English editing company to improve the language of the manuscript.

**6. Reviewer's comment:** Based on my above judgements, I suggest a major revision for the manuscript.

**Author's response:**

Thanks for your valuable comments and suggestions. We have carefully revised the full text, please pay attention to the revised manuscript.

**7. Reviewer's comment:** There are also lots of language and/or grammar and other minor issues. I just name a few:

**7.1 Reviewer's comment:** Line27-28, give references

**Author's response and changes in manuscript:**

Thanks for your suggestions. We have reorganized the language there and added references as follows.

*Having experienced the strong Cenozoic deformation, crust of this area develops a complex fault system with several large and deep faults, such as the generalized Haiyuan fault (F1), West Ordos fault (F2), West Qinling fault (F3), East Kunlun fault (F4), that divide the NETP into the Alxa, Ordos, Qilian, Qaidam, and Bayan Har blocks (Zhang et al., 2003; Fig.1). These faults are characterized by extremely intense tectonic movements and seismic activities (Zhang, 1999; Zheng et al., 2016b).*

**7.2 Reviewer's comment:** Line34, please explicitly indicate the earthquake locations in figure 1.

**Author's response and changes in manuscript:**

We have labeled all earthquakes greater than magnitude 8 on Fig. 1 as follows.

Figure 1. Earthquakes with magnitude M≥ 8.0 are labeled.

**7.3 Reviewer's comment:** Line36, what is the strength of an earthquake?

**Author's response and changes in manuscript:**

We have rephrased the sentence as follows.

*Since the generation and magnitude of an earthquake is closely related to fault activity, long-term fault slip rate plays a key role in medium- and long-term seismic hazard assessment (Ding et al., 1993; Xu et al., 2018).*

**7.4 Reviewer's comment:** Line39, the sentence reads quite strange, rephrase

**Author's response and changes in manuscript:**

We have rephrased the sentence as follows.

*For example, combined with coseismic displacements, long-term fault slip rates can be used to calculate earthquake recurrence interval (Shen et al., 2009) and assess the magnitudes of potential earthquakes (Bai et al., 2018; Hergert and Heidbach, 2010).*

**7.5 Reviewer's comment:** Line48-49, actually, quite a lot previous studies adopted elastic block models (e.g., Y. Li et al., 2017, 2021), and their results show non-negligible internal deformation

**Author's response and changes in manuscript:**

Thanks for pointing this out. We have rephrased the sentences as follows.

*For example, the geological slip rates only represent the activities of one fault branch that measured in a fault zone, which is always consist of several branches. They are usually lower than the geodetic slip rates on the fault as a whole if a rigid block assumption is adopted in the geodetic inversion process (Shen et al., 2009). However, several crustal deformation studies conducted in TP demonstrated that the internal block deformation in the NETP cannot be ignored (Royden et al., 1997; Zhang et al., 2004; Y. Li et al., 2017, 2021).*

**7.6 Reviewer's comment:** Line58, partitioning of deformation modes? What does it mean?

**Author's response and changes in manuscript:**

In the original manuscript, we intended to analyze whether the deformation pattern of the NETP is a block model or a continuum model according to the distribution characteristics of the velocity gradient. In revised manuscript, we will make adjustments to this description as follows.

*Based on these results, we summarized the long-term crustal deformation characteristics in the NETP.*

**7.7 Reviewer's comment:** Figure1, indicate the time span of earthquakes and the sources, give descriptions of P1-P3.

**Author's response and changes in manuscript:**

Thanks for your suggestions. We updated the Figure 1 caption as follows:

*Figure 1. Map of active faults and earthquakes of the NETP. Black lines represent the active faults. The light blue, red dots and the white pentagrams represent earthquakes from 1831 BC to 2017 AD from the National Earthquake Data Center (http://data.earthquake.cn). Black crosses (P1-P4) indicate the locations of four test sites for the comparison with the numerical model shown in Fig. 3b. Faults discussed in the text are labeled as followed…*

**7.8 Reviewer's comment:** Line84-85, cite references

**Author's response and changes in manuscript:**

We have added the corresponding references, and the revised text is shown below.

*Lithospheric faults (i.e., F1, F2, F3, and F4) cut through the Moho and reach the bottom of the model (Zhan et al., 2005; B. Liu et al., 2017; Zhao et al., 2015; Fig. 2a, b; Table 1). All other faults are crustal faults that terminate in the upper, middle, or lower crust (Yuan et al., 2002b, 2003; Lease et al., 2012; Meng et al., 2012; B. Liu et al., 2017; Wu et al., 2020).*

**7.9 Reviewer's comment:** Table1, list the references in the table

**Author's response and changes in manuscript:**

We added a column to the right of the table for references.

**Table 1.** Geometric parameters of faults in the model

|  | Fault name | Strike | Dip direction | Dip (°) | Reference |
|---|---|---|---|---|---|
| F1 | / | NWW-SSE | SW | 70 | RGAFSAO, 1988 |
| F2-1 | ZZSF | N-S | W | 70 | Gao, 2020 |
| F2-2 | HHF | N-S | W | 70 | Bao et al., 2019 |
| F2-3 | LSF | N-S | W | 80 | Wang et al., 2013 |
| F2-4 | YWSF | N-S | W | 70 | NIGS, 2017 |
| F2-5 | XGSF | N-S | W | 70 | NIGS, 2017 |
| F3-1 | DTH-LXF | NWW-SSE | NE | 70 | Zhou et al., 2009 |
| F3-2 | WQLF | | | | Li, 2005 |
| F4-1 | EKLF | NW-SE | NE | 75 | Z. Liu et al., 2017 |
| F4-2 | TZF | | | | J. Li et al., 2019 |
| F5 | East TJSF | NW-SE | SW | 70 | RGAFSAO, 1988 |
| | West TJSF | E-W | S | 70 | RGAFSAO, 1988 |
| F6 | WHLSF | N-S | W | 80 | Lei, 2015 |
| F7 | NSSF | NW-SE | SW | 70 | RGAFSAO, 1988 |
| F8 | EHLSF | NE-SW | SE | 60 | Du, 2010 |
| F9 | ZYGF | E-W | S | 60 | NIGS, 2017 |
| F10 | LHTF | NNE-SSW | SE | 70 | NIGS, 2017 |
| F11 | YCF | NNE-SSW | NW | 70 | NIGS, 2017 |
| F12 | YTSF | NW-SE | SW | 65 | NIGS, 2017 |
| F13 | QSHF | NW-SE | SW | 45 | Tian et al., 2020 |
| F14-1 | ZLHF | NNW-SSE | SW | 45 | Xu et al., 2016 |
| F14-2 | MXSF | NW-SE | SW | 80 | Hou et al., 1999 |
| F15 | LJSF | NWW-SSE | SW | 50 | Yuan et al., 2005 |
| F16-1 | DB-BLJF | NW-SE | SW | 70 | Yuan et al., 2007 |
| F16-2 | WD-KXF | E-W | SW | 70 | Jia et al., 2012 |

The detailed fault names are defined in Fig. 1.

**7.10 Reviewer's comment:** Line125, change critically important to important

**Author's response and changes in manuscript:**

Thanks for your suggestion. We have changed it in the revised manuscript.

**7.11 Reviewer's comment:** Line134, fitting misfit in mm/a or cm/a or others?

**Author's response:**

The *misfit* is a dimensionless unit. It was calculated according to the following formula (Cianetti et al., 2001):

$$misfit = \frac{\sum_i |\vec{V}_{GPS} - \vec{V}_{mod}|}{\sum_i |\vec{V}_{GPS}| + \sum_i |\vec{V}_{mod}|}$$

**7.12 Reviewer's comment:** Line135, I am not fully understood, why F2 and F3 are not considered in friction coefficients adjustments?

**Author's response:**

I guess you probably meant to refer to F2 and F4?

Actually, F1–F4 were all considered in the numerical simulation tests of friction coefficients adjustments. However, the adjustment of friction coefficients of F2 and F4 did little help to reduce the misfit value. Therefore, the friction coefficients of F2 and F4 remained unchanged.

**7.13 Reviewer's comment:** Line136, it seems to me, for F3, friction coefficient from 0.02 to 0.1 is large, why?

**Author's response:**

Let's first correct a mistake in the *misfit* calculation in Fig. 3a of the original manuscript. Due to a coding error, the calculation of *misfit* in the original manuscript only considers the easting components of the GPS observations and modeled velocities. Now we have corrected the error and the updated Fig. 3a is shown as follows. The value of 0.02 can still be considered as the best friction coefficient.

[Figure]

Figure 3. A friction coefficient of 0.02 or 0.03 yields the smallest fitting error.

Table R1. The simulation tests to find the lowest misfit

| $\mu'$ | F3=0.04 | F3=0.05 | F3=0.06 | F3=0.07 |
|---|---|---|---|---|
| *misfit* | 0.6078 | 0.05965 | 0.05855 | 0.05749 |
| $\mu'$ | F3=0.07; F1=0.01 | F3=0.08; F1=0.01 | F3=0.09; F1=0.01 | F3=0.10; F1=0.01 |
| *misfit* | 0.05639 | 0.05542 | 0.05449 | 0.05362 |

Now let's answer your question about "Line136, it seems to me, for F3, friction coefficient from 0.02 to 0.1 is large, why".

We determined this value through multiple numerical simulation tests. We found that increasing the friction coefficient of F3 and decreasing the friction coefficient of F1 is beneficial to the reduction of misfit. The simulation tests are shown as Table R1. The misfit value is the lowest when the friction coefficients of F1 and F3 are 0.01 and 0.1, respectively.

**7.14 Reviewer's comment:** Line163, Wang et al. (2020) should be Wang and Shen (2020), check the whole manuscript to avoid similar mistakes.

**Author's response and changes in manuscript:**

Thanks for pointing this out. We have corrected all similar mistakes.

**7.15 Reviewer's comment:** Line180, see the 5th major comment. I don't agree with this interpretation, velocity gradient within blocks might be related to block rotation as well!

**Author's response and changes in manuscript:**

Thanks for pointing this out. We have removed the relevant sentences.

**7.16 Reviewer's comment:** Line187, crustal velocity, not crust speed. Check and replace the whole manuscript

**Author's response and changes in manuscript:**

Thanks for pointing this out. We have checked the whole manuscript and made changes.

**7.17 Reviewer's comment:** Line215, older?

**Author's response and changes in manuscript:**

Thanks for pointing this out. We have replaced it with "earlier".

**7.18 Reviewer's comment:** Line215-217, give reasons for the discrepancies

**Author's response and changes in manuscript:**

We believe that there are two reasons for this.

First, early studies were based on geological methods with larger time scale. Second, the slip rate of the fault could not be better constrained in the past due to limited data (Li, et al., 2009).

We have already added it to the revised manuscript.

**7.19 Reviewer's comment:** Table3, change table to figure, which shows 1:1 plots of fault slip rates

**Author's response and changes in manuscript:**

Thanks for your suggestion.

We have carefully considered this issue, referring to the Fig. 3 in the paper of Y. Li et al. (2021). The difference between the results obtained by the two methods can be clearly intuitively observed from the figure.

However, the modeled slip rates vary with the locations of the faults in this paper. It is also inappropriate to use a mean value to replace the slip rate of the entire fault. Therefore, we retained the Table3 used in the original manuscript.

**7.20 Reviewer's comment: Line297, rephase the sentence**

**Author's response and changes in manuscript:**

Thanks for your suggestion. We have rephased the sentence as follows.

*In order to further examine the fit between the model results and GPS data, we selected a NE–SW profile that crosses through the study area (Fig. 11, C–C') and projected all GPS-observed values within 50 km of both sides of the profile.*

**7.21 Reviewer's comment:** Line300, see the 1st and 2nd major comments. It seems to me that the differences are large, especially across faults. The modeled velocities have steps across faults, this should be result from the fact that numerical model does not consider fault locking

**Author's response:**

Please see the responses to comment 2 and 3.

**7.22 Reviewer's comment:** Line315-319, I don't think the way of earthquake potential assessment appropriate. First, aseismic creep is not found along the Jinqianghe and Maomaoshan faults, as recent studies show. Second, the seismogenic does not corresponds to 20 km. Check the latest studies (e.g., Y. Li, 2021, JGR) to update your way of calculation.

**Author's response:**

Thanks for pointing this out.

The aseismic creep rates have been updated according to the latest study (Y. Li, et al., 2021). We also learned that the locking depth of the Laohushan fault or the Tianzhu Seismic Gap is about 20–22 km according to Y. Li et al. (2017, 2021), which is approximately equal to the data used in our manuscript. The new calculation results are shown in the table below.

[revised manuscript text omitted]

**Response to Comments from Reviewer #2**

Note that Section 3.2.4 of the original manuscript has been removed in this revision because we think that the F4 fault is not the primary research goal. Therefore, the sequence numbers of some figures may be changed in the revised manuscript.

**Reviewer's comment:** The article titled "Numerical Simulation of Present-day Kinematics at the Northeastern Margin of the Tibetan Plateau" focuses on the slip rates of active faults at the northeastern margin of the Tibetan Plateau. It is very important to understand the lateral expansion of the Tibetan Plateau and assess the seismic hazards in this region. By use of a three-dimensional geomechanics-numerical model, the authors obtained the horizontal and vertical crustal velocities and slip rates of active faults in the study area. The results are closely consistent with the observation in the area. The reviewer considers that this article is worth publishing on Solid Earth. The authors collected and researched previous relative works in this area and constructed a reasonable numerical model. The design for the geometric layering and block division, the rock properties, the boundary condition and other parameters are suitable. The results are reasonable and important for scientific research and seismic hazard analysis in this area. It is a pity that the authors have not paid attention to the important information given in Fig.11 during they discuss the "Fault slip rates and seismic hazards" in 4.2. Theoretically research and practical observations show that the isolated uplift area is the most dangerous place for earthquakes. So the review suggests that the authors pay attention to the result of the vertical velocity and slip rate of active faults in Fig. 11. The intersection location of areas with positive velocity value and the relative active fault is the most hazardous seismic location. The reviewer strongly suggests the authors reevaluate the earthquake risk regions by consideration of this factor combined with others. Another suggestion is that the authors can illustrate what finite element software is used in the modeling and the reason. Generally speaking, this is a very good article both in the modeling and in the research area.

**Author's response and changes in manuscript:**

Thanks for your constructive comments.
We have added a section to the revised manuscript to discuss the relationship between isolated uplift areas and earthquake occurrence, as follows.

*4.2.4 Isolated uplift areas and earthquakes*
*As mentioned above, we considered that earthquakes are less likely to occur on the Laohushan, Liupanshan and Haiyuan faults in the short term from the perspective of the earthquake recurrence cycle and the elapsed from the previous earthquake. However, the Haiyuan, Liupanshan, Lajishan and Daotanghe-Linxia faults are all located near the isolated rapid uplift areas of Qilian block (Fig. 10a). Many studies have also found that low-velocity bodies are widely distributed in the middle-lower*

*crust of the Qilian block (Bao et al., 2013; Wang et al., 2018; Ye et al., 2016). The spatial coupling of active faults, isolated uplift areas and low-velocity bodies is highly similar to the seismogenic conditions elaborated by the "seismic source cavity" model recently proposed by Zeng et al. (2021). That is, during the rapid uplift of the isolated areas (Fig. 10a), the low-velocity bodies in the middle-lower crust easily intruded into the weak space of the crust under the action of differential pressure to form a "seismic source cavity". If the isolated uplift areas keep to rise, the "seismic source cavity" may rise to the shallow part of the crust to intersect with brittle faults, causing strong earthquakes (Yang et al.,2009; Zeng et al., 2021). Therefore, in addition to the Jinqianghe and Maomaoshan faults mentioned above, the Haiyuan fault, Liupanshan fault, Lajishan fault and the Daotanghe-Linxia fault also have favorable structural conditions for strong earthquakes although some areas have not experienced in history.*

We also added a line to Section 2.5 describing the finite element software we used. The relevant text is as follows.

*For the calculation, we used the finite-element software Abaqus$^{TM}$ because its powerful nonlinear processing capabilities.*

**Response to Comments from Editor**

Note that Section 3.2.4 of the original manuscript has been removed in this revision because we think that the F4 fault is not the primary research goal. Therefore, the sequence numbers of some figures may be changed in the revised manuscript.

**1. Editor's comment:**
The paper is interesting, its structure is sound, but the grammar and wording are rather poor.

Concerning the content, I struggle to understand the purpose of the modeling. Since you have access to detailed GPS data, the measured velocity field (Fig.4) can be interpolated onto each fault plane to produce your fig.6 to fig.9. Why do you need to model the velocity field, why can't you use the GPS data?

If you are after fault properties, then these properties could be determined by minimizing the mismatch between the GPS data (Fig.4) and the modeled velocity field (Fig.5). But this would be a different paper.

**Author's response:**

First of all we need to correct a critical wording error in the title of the original manuscript. The modeled velocities in this paper refer to long-term velocities over several seismic cycles. We mistakenly used the word "present-day" in the title of the original manuscript and we have changed it to "contemporary".

The faults in this model are described by a pair of slidable contact surfaces. The mesh division of the contact surfaces is the same, so there are two nodes at any position of the fault, which belong to the two contact surfaces respectively. Driven by boundary conditions, the paired contact surfaces of each fault in the model undergo relative motion. According to the velocity of the fault nodes output by the model, the velocity difference of the paired nodes on the paired contact surfaces of the fault can be calculated one by one, so as to obtain the lateral slip rate of the fault. Take Fig. 6 as an example. Shown in the Fig.6 is the relative slip rate of the two contact surfaces of the fault, which cannot be obtained by GPS interpolation.

Moreover, the internal deformation of the blocks cannot be deducted by GPS interpolation to the fault plane, resulting in a high velocity on the fault plane. In addition, the small number of GPS stations may exacerbate the inaccuracy of the interpolation results.

It is also noted that we have used the services of a professional English editing company to improve the language of the manuscript. Hopefully, our fully revised version can meet the journal publication requirements.

**2. Editor's comment:**

Line2: Your title is not very informative. It states what you have been doing (numerical simulation of kinematics) but says nothing about the outcome of your work, which should be the focus of your title.

**Author's response and changes in manuscript:**

Thank you for this suggestion.
We changed the title to "Numerical Simulation of Contemporary Kinematics at the Northeastern Tibetan Plateau and its implications for seismic hazard assessment". It better reflects the work we have done and the corresponding outcomes.

**3. Editor's comment:**

Line 25–35: Odd sentence, rephrase; Vague, be more specific; Useless sentence…

**Author's response and changes in manuscript:**

Line 25-35 have been rephrased as follows:

*The northeastern Tibetan Plateau (NETP) is the growth front of the Tibetan Plateau (TP). Modern geomorphology and tectonic features of the NETP are inferred to be formed due to the expansion of the TP toward its periphery, which has been ongoing since the Indian and Eurasian plates collided (P. Zhang et al., 2013, 2014). Having experienced the strong Cenozoic deformation, crust of this area develops a complex fault system with several large and deep faults, such as the generalized Haiyuan fault (F1), West Ordos fault (F2), West Qinling fault (F3), East Kunlun fault (F4), that divide the NETP into the Alxa, Ordos, Qilian, Qaidam, and Bayan Har blocks (Zhang et al., 2003; Fig.1). These faults are characterized by extremely intense tectonic movements and seismic activities (Zhang, 1999; Zheng et al., 2016b). At least 5 earthquakes with magnitudes of ≥ 8, such as the 1654 M 8.0 Tianshui, 1739 M 8.0 Pingluo, 1879 M 8.0 Wudu, 1920 M 8.0 Haiyuan, and 1927 M 8.0 Gulang earthquakes, occurred in this area and caused huge loss of life and property in history (Fig. 1).*

**4. Editor's comment:**

Line 38: "Accurate fault slip rates can calculate seismic cycles (Shen et al., 2009) and assess the seismogenic potential ... ". Fault slip rates, accurate or not, can't calculate anything. Please re-write.

**Author's response and changes in manuscript:**

We have rephrased the sentences as follows.

*Since the generation and magnitude of an earthquake is closely related to fault activity, long-term fault slip rate plays a key role in medium- and long-term seismic hazard assessment (Ding et al., 1993; Xu et al., 2018). For example, combined with coseismic displacements, long-term fault slip rates can be used to calculate earthquake*

*recurrence interval (Shen et al., 2009) and assess the magnitudes of potential earthquakes (Bai et al., 2018; Hergert and Heidbach, 2010).*

**5. Editor's comment:**
All approaches have limitations and advantages. By combining several approaches, one can mitigate their respective limitation. What are the limitations and advantages of numerical simulations?

**Author's response and changes in manuscript:**

In this paper, one of the advantages of the numerical simulation is that we can obtain the 3D continuous slip rate of the faults. However, the modeled results strongly depend on the accuracy of the model input parameters. If we want to get reliable conclusions, we must set detailed parameters for the model, including model geometry, petrophysical properties, fault friction coefficient, initial crustal stress, and boundary conditions driving the model. Previous work on the NETP did not take these factors into account comprehensively, resulting in questionable results. The relevant text is in line 52–60 of the revised manuscript. The excerpt is as follows.

*Numerical modeling provides a powerful tool to study the large-scale crustal kinematics (Hergert and Heidbach, 2010; Hergert et al., 2011) as well as the comprehensive 3D view of fault activities with spatially continuous distribution. High efficiency and accuracy have made the numerical modeling a widespread technology in the field of geosciences, especially for the study of kinematics and dynamics of the NETP (Pang et al., 2019a, b; Sun et al., 2018, 2019; Zhu et al., 2018; Xiao and He, 2015). However, all these previous numerical models are either two-dimensional (2D) or three-dimensional (3D) with extremely simplified fault planes. To our knowledge, so far there is no 3D geomechanical model that take into the complex 3D fault system in the NETP. Therefore, detailed kinematics of the crust and faults in the NETP still remains unclear.*

**6. Editor's comment:**
Line 52: "as they provide a comprehensive view of current fault activities"
Not sure about this, can you please elaborate?

**Author's response and changes in manuscript:**

The results obtained by the geological method only represent the slip rate at one measurement point of the fault. Through numerical simulation, we can obtain the 3D motion state of any point of the entire fault plane from the model, as shown in Fig.6 to Fig.8. Line 52 have been rephrased as follows.

*Numerical modeling provides a powerful tool to study the large-scale crustal kinematics (Hergert and Heidbach, 2010; Hergert et al., 2011) as well as the comprehensive 3D view of fault activities with spatially continuous distribution.*

**7. Editor's comment:**

A proper Introduction should mention the main results.

**Author's response and changes in manuscript:**

Thanks for your suggestions. We have updated the last paragraph of Introduction as follows.

*In this study, instead of a simple conceptual model, a comprehensive 3D geomechanical model of the NETP with detailed complex 3D fault geometries, heterogeneous rock properties and reasonable initial crustal stress is constructed. After calibrated by model-independent observations, the results of the geomechannical model, such as the horizontal crustal velocities, spatially continuous slip rates of major faults, are presented. Based on these results, we summarized the long-term crustal deformation characteristics in the NETP. Finally, we assessed the seismic hazards of major faults in the study area, and suggested that the Jinqiangshan–Maomaoshan fault has the potential for a $M_S$ 7.1–7.3 earthquake in the coming decades.*

**8. Editor's comment:**

Line 79: "30 arcseconds". Give an indication in meter.

**Author's response and changes in manuscript:**

Thanks for your suggestion. We have added an indication in meter as follows.

*The topography of the model's surface is based on GTOPO30 elevation data, which has a resolution of 30 arcseconds (about 900m).*

**9. Editor's comment:**

Line 83–84: "... they cut through the Moho and reach the bottom of the model ..." How do you know? Any references to support this claim?

**Author's response and changes in manuscript:**

We have updated the relevant text as follows.

*Based on their depth, the faults of the model can be categorized into lithospheric and crustal faults. Lithospheric faults (i.e., F1, F2, F3, and F4) cut through the Moho and reach the bottom of the model (Zhan et al., 2005; B. Liu et al., 2017; Zhao et al., 2015; Fig. 2a, b; Table 1). All other faults are crustal faults that terminate in the upper, middle,*

*or lower crust (Yuan et al., 2002b, 2003; Lease et al., 2012; Meng et al., 2012; B. Liu et al., 2017; Wu et al., 2020).*

**10. Editor's comment:**
Line 125: "The frictional relations of the fault surface are critically important for the kinematics of a fault." What do you mean by "frictional relations"?

**Author's response and changes in manuscript:**

We have rephrased the sentence as follows.

*The friction coefficient of the fault surface is important for the kinematics of a fault.*

**11. Editor's comment:**
Line 142: "which predicts that all deformations due to gravitational loading occur in the vertical direction and that no expansion or contraction occurs in the lateral direction." Does this means that there is no horizontal gravitational forces due to lateral variation of gravitational potential energy?

"Loading" suggests "stress", but "expansion" and "contraction" suggest strain. This is a bit confusing. Can you rephrase this by saying that vertical stress, leads to horizontal stress via the Poisson's ratio...

**Author's response and changes in manuscript:**

Yes, you are quite right form a global perspective of the model. However, in the local part of the model, there will be a horizontal gravitational force caused by the lateral variation of gravitational potential energy, so that the material is force-balanced.

The sentence has been phrased as follows.

*The initial stress state that is most commonly employed in previous numerical modeling studies of the NETP is the uniaxial strain reference state (Zhu et al., 2016), which based on the boundary condition that no elongation occurs in the horizontal direction, and the strain only occurs in the vertical direction.*

**12. Editor's comment:**

Line 149: "Furthermore, k-values obtained globally from in situ measurements always greatly exceed 1/3 (Hergert and Heidbach, 2011)."

Shouldn't it exceed 1? i.e. horizontal stress > vertical stress?

**Author's response:**

Global stress magnitude measurements show that the horizontal stress is generally greater than the vertical stress in the shallow crust, but this is not the case for all measured data, as shown in the Fig. R1. Therefore, "always greatly exceed 1/3" might be a more reasonable description.

[Figure]

Figure R1. Global compilation of stress magnitude measurements (Hergert and Heidbach, 2011).

**13. Editor's comment:**

Line 165: "it was assumed that the lateral velocities of the 3D model do not vary with depth". Why is this a reasonable assumption? and what would be the consequences if velocity were depth-dependent?

**Author's response:**

The observed vertically coherent deformation imply that the crust and lithospheric mantle are mechanically coupled (Wang et al., 2008). Thus it was assumed that the lateral velocities of the 3D model do not vary with depth. This assumption is also widely used in previous numerical simulation studies (Xiao and He, 2015; Li, Hergert et al., 2021; Sun and Luo, 2018).

**14. Editor's comment:**

Figure 4: Please add faults' id. The integration of the velocity field along the boundary should give the amount of material entering (overall thickenning) or leaving (overall thinning) the cartesian model.

**Author's response and changes in manuscript:**

The boundary condition we impose on the model is displacement, and the total amount of material in the model is constant. Therefore, there is no need to emphasize the material entering or leaving. The faults' id of Fig. 4 has been added as follows.

[Figure]

Figure 4 The faults' id has been added.

**15. Editor's comment:**

Line 180: "indicating that their internal deformation is low". I don't understand the logic of this statement. The velocity high or low says nothing about internal strain (i.e., a rigid block could move very fast without internal deformation). Gradients of velocity in the other hand indicates internal strain.

**Author's response and changes in manuscript:**

Thanks for your comment. The statement is indeed incorrect. We have removed the sentence in the revised manuscript.

**16. Editor's comment:**
Figure 5: Can you plot the mismatch between figure 4 (observed velocity) and figure 5 (calculated velocity)?

**Author's response and changes in manuscript:**

Thanks for your suggestion. Actually, the comparison between observed velocity and calculated velocity has been plotted in Fig. 11 of the revised manuscript.

**17. Editor's comment:**
Figure6: Only panel a/ is useful.

**Author's response and changes in manuscript:**

We believe that these subgraphs can reflect the motion state of the fault more concretely.

Although subgraphs of (b), (c) and (d) can be considered subsets of subgraph (a), the specific information contained in them may also be of interest to some readers.

**18. Editor's comment:**
Line 206–208: "Can you please locate these earthquakes on Figure 4?"

**Author's response and changes in manuscript:**

Thanks for your suggestion.
Figure 4 integrates less information and is only used to show the boundary conditions of the model. We have labeled strong earthquakes with magnitudes M ≥ 8.0 in Figure 1 as follows.

[Figure]

Figure 1. Earthquakes with magnitude M≥ 8.0 are labeled.

**19. Editor's comment:**

Line 219, Line221: "Liupanshan faults have the rake ranging from 10° to 20°"
A fault plane has no rake. A striae on a fault plane has a rake (rake = 90 - the striae pitch). It looks like your rake is in fact the pitch.

**Author's response:**

As we replied in the comment 1, the faults in this model are described by a pair of slidable contact surfaces. Driven by boundary conditions, the paired contact surfaces of each fault in the model undergo relative motion. The slip rakes in line 219 and 221 mean the slip directions on the surfaces. We used GeoStress to calculate it (Stromeyer et al., 2020).

**20. Editor's comment:**

Figure 10: Can you compare and contrast with the observed slip rates and slip senses?

**Author's response:**

We believe that there has been a misunderstanding about the slip rates in Fig. 10 of the original manuscript.

Plotted in Fig.10 (Fig. 9 in the revised manscript) is the long-term slip rate of the fault over several seismic cycles which is totally different from the GPS observations obtained during the interseismic periods. There is no comparability between them.

**21. Editor's comment:**

Line 270: "Given the zero vertical velocity imposed at the base of the model, is the calculated vertical velocity field of any significance?"

**Author's response:**

The modeled vertical motion at the surface is hardly affected by the motion state of the bottom of the model. On the contrary, it is generated by the horizontal motion on a complex model that includes factors such as fault geometry, topography, crustal interfaces, etc.

Actually, there is a high consistency between the modeled vertical velocities at the surface and the basin subsidence rates obtained by geological means (Wang et al., 2011; Ma et al., 20221).

**22. Editor's comment:**

Figure 13: Can you do the same thing but along the faults planes? So you can compare observed and calculaled velocity on faults?

**Author's response:**

Thank you for bringing this to our attention.

However, the slip rate of a fault is obtained by calculating the relative motion of the paired contact surfaces. It is different from the GPS observations interpolated to the fault plane. Therefore, there is no comparability between them.

**23. Editor's comment:**

Table 3: Can you please explain the purpose of the modeling? It looks to me that the GPS data is sufficient to extrapolate the velocity field from which the velocity field on each fault can be determined. One could simply use the velocity field to constraint faults properties via a mismatch minimization procedure.

**Author's response:**

Please see the response to the comment 1.

**24. Editor's comment:**

Line 298: "we selected a SW–NE profile that covers the study area (Figure 12, C–C')
and projected all GPS-observed values within 50 km onto the profile".
Please do this on faults and compare with 6, 7 and 8.

**Author's response:**

As we reply to comment 22, the slip rate of a fault is obtained by calculating the relative
motion of the paired contact surfaces. It is different from the GPS observations
interpolated to the fault plane. Therefore, there is no comparability between them.

**25. Editor's comment:**

Line 307: The slip rates (2.6–3.0 mm/a) simulated for the F2-2 Luoshan Fault are
similar to the measured slip rate (2.2 mm/a) and the slip rates simulated for the West
Qinling Fault in the Zhangxian and Tianshui region (2.4–3.0 mm/a, Figure 10) are
consistent with the slip rates obtained using geological methods (2.5–2.9 mm/a; Chen
et al., 2019).
Can you quantify this a bit better. "Similar" is a bit vague.

**Author's response and changes in manuscript:**

The sentence has been rephrased as follows.

*The modeled slip rates on the F2-3 Luoshan Fault (2.6–3.0 mm/a) are in line with the
geological slip rate (2.2 mm/a). A good agreement between these two kinds of slip rates
also exists on the West Qinling Fault in the Zhangxian and Tianshui region (Table 3).*

**26. Editor's comment:**

Line 312: "The M8.0 Gulang earthquake occurred in 1927 in the northwestern part of
the F1 fault, whereas the M8.0 Haiyuan earthquake occurred in 1920 on the Haiyuan
Fault". Can you show these earthquakes on a figure?

**Author's response and changes in manuscript:**

We have labeled these earthquakes in Fig. 1 of the revised manuscript, as shown in
response to comment 18.

**27. Editor's comment:**

Line 317-321: "Based on the slip rates and other fault data, we estimated the earthquake magnitude based on the energy accumulated during elapsed time (Purcaru et al., 1978) and recurrence intervals (Shen et al., 2009), as shown in Table 4. The Jinqianghe, Maomaoshan, and Laohushan faults can generate $M_S$6.9, $M_S$7.2, and $M_S$6.8 earthquakes, with recurrence intervals of 320 707, 890, and 1132 years, respectively."
This seems an important result, can you detail the procedure?

**Author's response and changes in manuscript:**

We have updated the calculation according to the RC1's comment. The updated results are shown as follows.

**Table 4.** Earthquake magnitude and recurrence interval of each fault based on the energy accumulated during the elapsed time since the last remarkable earthquake

| Fault name | $V_1$(mm/a) | $V_2$(mm/a) | $L_1$ (km) | $L_2$ (km) | $\mu$ (Gpa) | t | S (m) | $M_S$ | T (a) |
|---|---|---|---|---|---|---|---|---|---|
| JQHF | 3.5 | / | 34 | 20 | 34.5 | 675 | 1.5 | 7.1 | 424 |
| MMSF | 3.9 | / | 51 | 20 | 34.5 | 952 | 2.2 | 7.3 | 571 |
| LHSF | 4.1 | 2.5 | 70 | 20 | 34.5 | 133 | 3.1 | 6.6 | 1910 |
| MSLPSF, SSLPSF | 2.5 | / | 80 | 23 | 34.5 | 570 | 3.5 | 7.2 | 1397 |
| GG-BJF | 0.7 | / | 70 | 23 | 34.5 | 1400 | 3.1 | 7.1 | 4365 |

$V_1$ is the modeled average slip rate of the fault in this study; $V_2$ is the aseismic creep rate of the fault (Y. Li, et al., 2021); $L_1$ is the length of the fault (Xu et al., 2016); $L_2$ is the depth of the seismogenic, which refers to the locking depth(Y. Li, et al., 2017, 2021); $\mu$ is the shear modulus of the rocks (Aki et al., 2002); t is the time that has elapsed since the most recent remarkable earthquake (Gan et al., 2002; Shi et al., 2013, 2014; Wang et al., 2001); S is the largest maximum coseismic displacement, calculated using the method of Gan et al. (2002); $M_S$ is the earthquake magnitude corresponding to the energy accumulated by the fault between recurrences (Purcaru et al., 1978); T is the recurrence interval of the earthquake, where $T = S/(V_1-V_2)$ (Shen et al., 2009). The fault names are defined in Fig. 1 and Fig. 13.

The magnitude of the earthquake ($M_S$) corresponding to the energy accumulated during the elapsed time can be estimated by the following formula (Purcaru et al., 1978):

$$M_0 = \mu AD$$
$$\log M_0 = 1.5M_s + 9.1$$

where $M_0$ is the seismic moment (N.m), $\mu$ is the shear modulus of the rock, and A is the rupture area of the fault (A = $L_1$ * $L_2$), D is the average displacement of fault during the elapsed time (D = ($V_1 - V_2$) * t). S is calculated by empirical formula in the TP ($lgS = -1.36 + lgL1$; Gan et al., 2002). T is the recurrence interval of the fault, where T = S/($V_1$-$V_2$) (Shen et al., 2009).

**28. Editor's comment:**

Figure 14 (Figure 13 in the revised manuscript):
There figures are confusing, please add an arrow to clearly show the north direction.

**Author's response and changes in manuscript:**

Thanks for your suggestion, we have updated the figures as follows.

[Figure]

Figure 13 Arrows were added to show the north direction.

**29. Editor's comment:**

Line 340: "by three sources of tectonic stress".

Very poor working. There are three interacting blocks, separated by fault zones, each with contrasting velocity fields.

**Author's response and changes in manuscript:**

We have rephrased the sentence as follows.

*The Liupanshan and Guguan–Baoji faults are jointly affected by three interacting blocks with contrasting velocity fields (Fig. 13b)*

**30. Editor's comment:**

Line 344-345: "Third, our simulations show that the Yunwushan–Xiaoguanshan Fault has a significant right-lateral strike-slip component".

I am pretty sure that this can be seen in the field. Why rely on numerical modeling, when field geology can provide observables?

**Author's response:**

The continuous slip rates of the fault can be obtained by numerical modeling, which is almost impossible in field work. Actually, few literatures are devoted to the study of the Yunwushan-Xiaoguanshan fault, whether through field work or other means.

**31. Editor's comment:**

Line 364: "we estimated that the energy accumulated…"

How was this estimation made? Or is it just a speculation?

**Author's response:**

Please see the reply to the comment 27.

**32. Editor's comment:**

Line 366: "Because the next event is not likely to occur for a long time, the…"

This is a rather bold statement to make. What if a Ms7 earthquake happens in the next few years? Could the authors' responsibility be engaged, and could they be liable?

**Author's response and changes in manuscript:**

Thanks for your suggestion. We have rephrased the sentences as follows.

*Therefore, we infer that the middle-southern Liupanshan fault and the Guguan-Baoji fault are most likely in a state of stress accumulation, and the likelihood of a large earthquake on these fault segments in the next few decades is thought to be low.*

**32. Editor's comment:**

Line 377: "the left-lateral strike-slip motions of the Maxianshan Fault may not be as intense as previously thought".

What does "intense motion" mean?

**Author's response and changes in manuscript:**

We have rephrased the sentence as follows.

*the left-lateral strike-slip rates of the Maxianshan Fault may not be as large as previously thought.*

**33. Editor's comment:**

Line 395: "This implies that the deformations of the TP should display "aggregated" 395 characteristics."

Not sure what this means "aggregated characteristics"? Can you please rephrase?

**Author's response and changes in manuscript:**

Based on the comments of the RC1, we have rewritten Section 4.3. This sentence has been removed in the revised manuscript.

**References** (only new)

Lease R. O., Burbank D. W., Zhang H., Liu J., and Yuan D.: Cenozoic shortening budget for the northeastern edge of the Tibetan Plateau: Is lower crustal flow necessary? , Tectonics, 31, TC3011, https://doi.org/10.1029/2011TC003066, 2012.

Li Y., Nocquet J. M., Shan X., and Song X.: Geodetic observations of shallow creep on the Laohushan-Haiyuan Fault, Northeastern Tibet, J. Geophys. Res. Sol. Ea., 126, https://doi.org/10.1029/2020JB021576, 2021.

Liu B., Feng S., Ji J., Wang S., Zhang J., Yuan H., and Yang G.: Lithospheric structure and faulting characteristics of the Helan Mountains and Yinchuan Basin: Results of deep seismic reflection profiling, Sci. China Earth Sci., 60, 589–601, https://doi.org/10.1007/s11430-016-5069-4, 2017.

Meng X., Shi L., Guo L., Tong T., and Zhang S. Multi-scale analyses of transverse structures based on gravity anomalies in the northeastern margin of the Tibetan Plateau, Chinese J. Geophys., https://doi.org/10.6038/j.issn.0001-5733.2012.12.006, 2012.

Wang C., Flesch L. M., Silver P. G., Chang L., and Chan W. W.: Evidence for mechanically coupled lithosphere in central Asia and resulting implications, Geology, 36, 363–366, https://doi.org/10.1130/G24450A.1, 2008.

Wu G., Tan H., Sun K., Wang J., Xi Y., and Shen C.: Characteristics and tectonic significance of gravity anomalies in the Helanshan-Yinchuan Graben and adjacent areas, Chinese J., Geophys., 63, 1002–1013, https://doi.org/10.6038/cjg2020N0233, 2020.

Yuan D., Liu B., Zhang P., Liu X., Cai S., and Liu X.: The neotectonic deformation and earthquake activity in Zhuanglang river active fault zone of Lanzhou, Acta Seismologica Sinica, 24, 441–444, 2002b.

Zhan Y., Zhao G., Wang J., Tang J., Chen X., Deng Q., Xuan F., and Zhao J.: Crustal electric structure of Haiyuan arcuate tectonic region in the northeastern margin of Qinghai-Xizang Plateau, China. Acta Seismologica Sinica, 27, 431–440, 2005.

Zheng W., Yuan D., Zhang P., Yu J., Lei Q., Wang W., Zheng D., Zhang H., Li X., Li C., and Liu X.: Tectonic geometry and kinematic dissipation of the active faults in the northeastern Tibetan Plateau and their implications for understanding northeastward growth of the plateau, Quaternary Sciences, 36, 775–788, https://doi.org/10.11928/j.issn.1001-7410.2016.04.01, 2016b.

Zhao L., Zhan Y., Chen X., Yang H., and Jiang F.: Deep electrical structure of the central West Qinling orogenic belt and blocks on its either side, Chinese J. Geophys., 58, 2460–2472, https://doi.org/10.6038/cjg20150722, 2015.

---

## Author Response (AR2)

July 8, 2022

Patrice Rey
Topical editor
*Solid Earth*

Dear Editor,
Thank you for the helpful comments. We have considered all the comments and revised the manuscript accordingly. It is also noted that we have used the services of a professional English editing company to improve the language of the manuscript. We have attached an editing certificate below.

Please find your point-by-point response to the comments below. We hope that the revised manuscript is now ready for publication in *Solid Earth*.

Thank you again for your consideration.

Sincerely,
Liming Li
School of Earth Sciences and Resources,
China University of Geosciences (Beijing),
No.29, Xueyuan Road, Haidian District, Beijing, P.R. China
Email: cugliming@cug.edu.cn

[Figure]

**Editing Certificate**

This document certifies that the manuscript listed below has been edited to ensure language and grammar accuracy and is error free in these aspects. The logical presentation of ideas and the structure of the paper were also checked during the editing process. The edit was performed by professional editors at Editage, a division of Cactus Communications. The sections titled references were not edited by Editage upon the author's request.

The author's core research ideas were not altered during the editing process. Editage guarantees the quality of editing with the assumption that our suggested changes have been accepted and the edited text has not been altered without the knowledge of our editors.

**MANUSCRIPT TITLE**

**Numerical Simulation of Contemporary Kinematics at the Northeastern Tibetan Plateau and its implications for seismic hazard assessment**

**AUTHORS**

**Liming Li**

**ISSUED ON**

**June 30, 2022**

**JOB CODE**

**ILIMI_4_4**

[Figure]

[Figure]

**Vikas Narang**
**Chief Operating Officer - Editage**
* * *
**editage**

Editage, a brand of Cactus Communications, offers professional English language editing and publication support services to authors engaged in over 1300 areas of research. Through its community of experienced editors, which includes doctors, engineers, published scientists, and researchers with peer review experience, Editage has successfully helped authors get published in internationally reputed journals. Authors who work with Editage are guaranteed excellent language quality and timely delivery.

**GLOBAL :**
+1(833) 979-0061 | request@editage.com

**CHINA :**
400-120-3020 | fabiao@editage.cn

**CACTUS.**

IMPACT SCIENCE — impact.science

researcher.life

CACTUS LIFE SCIENCES — lifesciences.cactusglobal.com

**Response to Comments from Editor**

**1. Editor's comment:**
Introduction: Concerning the timing of the onset of the collision between India and Asia, there are older references than Zhang et al., 2013, 2014. Consider referencing also older papers such as Achache et al., 1984.

**Author's response:**

Thank you for your suggestion; we have added two references. The updated sentence is as follows:

*Modern geomorphology and tectonic features of the NETP are thought to be formed by the expansion of the TP toward its periphery, which has been ongoing since the initial collision of the Indian and Eurasian plates (Achache et al., 1984; Patriat and Achache 1984; P. Zhang et al., 2013, 2014).*

**2. Editor's comment:**
Line 28: remove "generalized".

**Author's response:**

Thank you for your suggestion. We have removed "generalized" in the revised manuscript.

**3. Editor's comment:**
Line 30: Remove "extremely".

**Author's response:**

Thank you for your suggestion. We have removed "extremely" in the revised manuscript.

**4. Editor's comment:**
Line 33: Remove "in history".

**Author's response:**

Thank you for your suggestion. We have removed "in history" in the revised manuscript.

**5. Editor's comment:**
Line 37: "... which are lack in the NETP ...", can you please correct this mistake?

**Author's response:**

Thank you for your suggestion. The sentence has been rephrased as follows:

*Moreover, the spatially continuous fault slip rates that NETP lacks can also be used to reconstruct the tectonic evolution of this area and provide important insights into the lateral expansion pattern and deformation mechanisms of the TP (Royden et al., 1997; Tapponnier et al., 1982; Zhang et al., 2004).*

**6. Editor's comment:**
Line 58: "... instead of a simple conceptual model ..." avoid this kind of unnecessary innuendo aimed at previous research. Your own model is still conceptual and very simple.

**Author's response:**

Thank you for your suggestion. The sentence has been rephrased as follows:

*In this study, a comprehensive 3D geomechanical model of the NETP was constructed with detailed complex 3D fault geometries, heterogeneous rock properties and reasonable initial crustal stress.*

**7. Editor's comment:**
Line 60: Replace "After calibrated" with "After calibration with..."

**Author's response:**

Thank you for your suggestion. The sentence has been rephrased as follows:

*After calibration with model-independent observations, the results of the geomechanical model, such as the horizontal crustal velocities and spatially continuous slip rates of major faults, were presented.*

**8. Editor's comment:**
Line 95: Replace "detachment" by "décollement". A detachment cuts across the rheological layering, a decollement runs parallel to it.

**Author's response:**

Thank you for your suggestion. We have replaced "detachment" with "decollement" in the revised manuscript.

9. **Editor's comment:**
Line 160: No need to capitalize "Where".

**Author's response:**

Thank you for your suggestion. This correction has been made.

In addition, we have made substantial language revisions. Please see the marked-up manuscript for more detail.